# PSA-GAN: Progressive Self Attention GANs for Synthetic Time Series

**Paul Jeha**[*,†]
Technical University
of Denmark
`pauje@dtu.dk`

**Michael
Bohlke-Schneider**[*]
AWS AI Labs
`bohlkem@amazon.com`

**Pedro Mercado**
AWS AI Labs
`pedroml@amazon.com`

**Shubham Kapoor**
AWS AI Labs
`kapooshu@amazon.com`

**Rajbir Singh Nirwan**
AWS AI Labs
`nirwar@amazon.com`

**Valentin Flunkert**
AWS AI Labs
`flunkert@amazon.com`

**Jan Gasthaus**
AWS AI Labs
`gasthaus@amazon.com`

**Tim Januschowski**[†]
Zalando SE
`tim.januschowski@zalando.de`

## Abstract

Realistic synthetic time series data of sufficient length enables practical applications in time series modeling tasks, such as forecasting, but remains a challenge. In this paper, we present `PSA-GAN`, a generative adversarial network (GAN) that generates long time series samples of high quality using progressive growing of GANs and self-attention. We show that `PSA-GAN` can be used to reduce the error in several downstream forecasting tasks over baselines that only use real data. We also introduce a Frechet Inception distance-like score for time series, Context-FID, assessing the quality of synthetic time series samples. We find that Context-FID is indicative for downstream performance. Therefore, Context-FID could be a useful tool to develop time series GAN models.

## 1 Introduction

In the past years, methods such as (Salinas et al., 2020; Franceschi et al., 2019; Kurle et al., 2020; de Bézenac et al., 2020; Oreshkin et al., 2020a; Rasul et al., 2021; Cui et al., 2016; Wang et al., 2017) have consistently showcased the effectiveness of deep learning in time series analysis tasks. Although these deep learning based methods are effective when sufficient and clean data is available, this assumption is not always met in practice. For example, sensor outages can cause gaps in IoT data, which might render the data unusable for machine learning applications (Zhang et al., 2019b). An additional problem is that time series panels often have insufficient size for forecasting tasks, leading to research in meta-learning for forecasting (Oreshkin et al., 2020b). Cold starts are another common problem in time series forecasting where some time series have little and no data (like a new product in a demand forecasting use case). Thus, designing flexible and task-independent models that generate synthetic, but realistic time series for arbitrary tasks are an important challenge. Generative adversarial networks (GAN) are a flexible model family that has had success in other domains. However, for their success to carry over to time series, synthetic time series data must be of realistic length, which current state-of-the-art synthetic time series models struggle to generate because they often rely on recurrent networks to capture temporal dynamics (Esteban et al., 2017; Yoon et al., 2019).

In this work, we make three contributions: i) We propose `PSA-GAN` a progressively growing, convolutional time series GAN model augmented with self-attention (Karras et al., 2017; Vaswani et al., 2017). `PSA-GAN` scales to long time series because the progressive growing architecture starts

---

[*]Equal contribution.
[†]Work done while at Amazon/AWS AI Labs.

modeling the coarse-grained time series features and moves towards modeling fine-grained details during training. The self-attention mechanism captures long-range dependencies in the data (Zhang et al., 2019a). ii) We show empirically that `PSA-GAN` samples are of sufficient quality and length to boost several downstream forecasting tasks: far-forecasting and data imputation during inference, data imputation of missing value stretches during training, forecasting under cold start conditions, and data augmentation. Furthermore, we show that `PSA-GAN` can be used as a forecasting model and has competitive performance when using the same context information as an established baseline. iii) Finally, we propose a Frechet Inception distance (FID)-like score (Salimans et al., 2016), Context-FID, leveraging unsupervised time series embeddings (Franceschi et al., 2019). We show that the lowest scoring models correspond to the best-performing models in our downstream tasks and that the Context-FID score correlates with the downstream forecasting performance of the GAN model (measured by normalized root mean squared error). Therefore, the Context-FID could be a useful general-purpose tool to select GAN models for downstream applications.

We structure this work as follows: We discuss the related work in Section 2 and introduce the model in Section 3. In Section 4, we evaluate our proposed GAN model using the proposed Context-FID score and through several downstream forecasting tasks. We also directly evaluate our model as a forecasting algorithm and perform an ablation study. Section 5 concludes this manuscript.

## 2 RELATED WORK

GANs (Goodfellow et al., 2014) are an active area of research (Karras et al., 2019; Yoon et al., 2019; Engel et al., 2019; Lin et al., 2017; Esteban et al., 2017; Brock et al., 2018) that have recently been applied to the time series domain (Esteban et al., 2017; Yoon et al., 2019) to synthesize data (Takahashi et al., 2019; Esteban et al., 2017), and to forecasting tasks (Wu et al., 2020). Many time series GAN architectures use recurrent networks to model temporal dynamics (Mogren, 2016; Esteban et al., 2017; Yoon et al., 2019). Modeling long-range dependencies and scaling recurrent networks to long sequences is inherently difficult and limits the application of time series GANs to short sequence lengths (less than 100 time steps) (Yoon et al., 2019; Esteban et al., 2017). One way to achieve longer realistic synthetic time series is by employing convolutional (van den Oord et al., 2016; Bai et al., 2018; Franceschi et al., 2019) and self-attention architectures (Vaswani et al., 2017). Convolutional architectures are able to learn relevant features from the raw time series data (van den Oord et al., 2016; Bai et al., 2018; Franceschi et al., 2019), but are ultimately limited to local receptive fields and can only capture long-range dependencies via many stacks of convolutional layers. Self-attention can bridge this gap and allow for modeling long-range dependencies from convolutional feature maps, which has been a successful approach in the image (Zhang et al., 2019a) and time series forecasting domain (Li et al., 2019; Wu et al., 2020). Another technique to achieve long sample sizes is progressive growing, which successively increases the resolution by adding layers to generator and discriminator during training (Karras et al., 2017). Our proposal, `PSA-GAN`, synthesizes progressive growing with convolution and self-attention into a novel architecture particularly geared towards time series.

Another line of work in the time series field is focused on developing suitable loss functions for modeling financial time series with GANs where specific challenges include heavy tailed distributions, volatility clustering, absence of autocorrelations, among others (Cont, 2001; Eckerli & Osterrieder, 2021). To this end, several models like QuantGAN (Wiese et al., 2020), (Conditional) SigWGAN (Ni et al., 2020; 2021), and DAT-GAN (Sun et al., 2020) have been proposed (see (Eckerli & Osterrieder, 2021) for review in this field). This line of work targets its own challenges by developing new loss functions for financial time series, which is orthogonal to our work, i.e. we focus on neural network architectures for time series GANs and show its usefulness in the context of time series forecasting.

Another challenge is the evaluation of synthetic data. While the computer vision domain uses standard scores like the Inception Score and the Frechet Inception distance (FID) (Salimans et al., 2016; Heusel et al., 2017), such universally accepted scores do not exist in the time series field. Thus, researchers rely on a *Train on Synthetic–Test on Real* setup and assess the quality of the synthetic time series in a downstream classification and/or prediction task (Esteban et al., 2017; Yoon et al., 2019). In this work, we build on this idea and assess the GAN models through downstream forecasting tasks. Additionally, we suggest a Frechet Inception distance-like score that is based on unsupervised time series embeddings (Franceschi et al., 2019). Critically, we want to be able to score the fit of our fixed

length synthetic samples into their context of (often much longer) true time series, which is taken into account by the contrastive training procedure in Franceschi et al. (2019). As we will later show, the lowest scoring models correspond to the best performing models in downstream tasks.

## 3   MODEL

**Problem formulation**   We denote the values of a time series dataset by $z_{i,t} \in \mathbb{R}$, where $i \in \{1, 2, \ldots, N\}$ is the index of the individual time series and $t \in \{1, 2, \ldots, T\}$ is the time index. Additionally, we consider an associated matrix of time feature vectors $\mathbf{X}_{1:T} = (\mathbf{x}_1, \ldots, \mathbf{x}_T)$ in $\mathbb{R}^{D \times T}$. Our goal is to model a time series of fixed length $\tau$, $\hat{Z}_{i,t,\tau} = (\hat{z}_{i,t}, \ldots, \hat{z}_{i,t+\tau-1})$, from this dataset using a conditional generator function $G$ and a fixed time point $t$. Thus, we aim to model $\hat{Z}_{i,t,\tau} = G(\mathbf{n}, \phi(i), \mathbf{X}_{t:t+\tau-1})$, where $\mathbf{n} \in \mathbb{R}^\tau$ is a noise vector drawn from a Gaussian distribution of mean zero and variance one; $\phi$ is an embedding function that maps the index of a time series to a vector representation, that is concatenated to each time step of $\mathbf{X}_{t:t+\tau-1}$. An overview of the model architecture is shown in Figure 1 and details about the time features are presented in Appendix A.

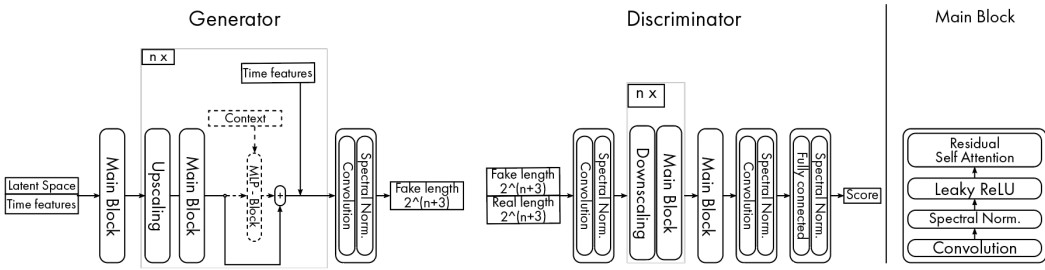

Figure 1: *Left*: Architecture of our proposed model, `PSA-GAN`. The generator contains $n$ blocks where each doubles the size of the output, by linear interpolation. It contains (dashed drawing) a multi layer perceptron block, that incorporates knowledge of the past in the generator. This block is used in the `PSA-GAN-C` model. The discriminator contains $n$ blocks that halve the size of the input using average pooling. *Right*: The main block used in the generator and discriminator.

**Spectral Normalised Residual Self-Attention with Convolution**   The generator and discriminator use a main function $m$ that is a composition of convolution, self-attention and spectral normalisation

$$m \circ f : \mathbb{R}^{n_f \times l} \to \mathbb{R}^{n_f \times l}$$
$$x \mapsto \gamma \operatorname{SA}(f(x)) + f(x) \tag{1}$$

where $f(x) = \operatorname{LR}(\operatorname{SN}(c(x)))$ and $m(y) = \gamma \operatorname{SA}(y) + y$, $c$ is a one dimensional convolution operator, LR the LeakyReLU operator (Xu et al., 2015), SN the spectral normalisation operator (Miyato et al., 2018) and SA the self-attention module. The variable $n_f$ is the number of in and out-channels of $c$, $l$ is the length of the sequence. Following the work of (Zhang et al., 2019a), the parameter $\gamma$ is learnable. It is initialized to zero to allow the network to learn local features directly from the building block $f$ and is later enriched with distant features as the absolute value of gamma increases, hereby more heavily factoring the self-attention term SA. The module $m$ is referenced as residual self-attention in Figure 1 (*right*).

**Downscaling and Upscaling**   The following sections mention upscaling (UP) and downscaling (DOWN) operators that double and halve the length of the time series, respectively. In this work, the upscaling operator is a linear interpolation and the downscaling operator is the average pooling.

**PSA-GAN**   `PSA-GAN` is a progressively growing GAN (Karras et al., 2017); thus, trainable modules are added during training. Hereby, we model the generator and discriminator as a composition of functions: $G = g_{L+1} \circ \ldots \circ g_1$ and $D = d_1 \circ \ldots \circ d_{L+1}$ where each function $g_i$ and $d_i$ for $i \in [1, L+1]$ corresponds to a module of the generator and discriminator.

GENERATOR    As a preprocessing step, we first map the concatenated input $[\mathbf{n}, \phi(i), \mathbf{X}_{t:t+\tau-1}]$ from a sequence of length $\tau$ to a sequence of length 8, denoted by $\tilde{Z}_0$, using average pooling. Then, the first layer of the generator $g_1$ applies the main function $m$:

$$g_1 : \mathbb{R}^{n_f \times 2^3} \rightarrow \mathbb{R}^{n_f \times 2^3}$$
$$\tilde{Z}_0 \mapsto \tilde{Z}_1 = m \circ f(\tilde{Z}_0) \tag{2}$$

For $i \in [2, L]$, $g_i$ maps an input sequence $\tilde{Z}_{i-1}$ to an output sequence $\tilde{Z}_i$ by applying an upscaling of the input sequence and the function $m \circ f$:

$$g_i : \mathbb{R}^{n_f \times 2^{i+1}} \rightarrow \mathbb{R}^{n_f \times 2^{i+2}}$$
$$\tilde{Z}_{i-1} \mapsto \tilde{Z}_i = m \circ f(\mathrm{UP}(\tilde{Z}_{i-1})) \tag{3}$$

The output of $g_i$ is concatenated back to the time features $\mathbf{X}_{t:t+\tau-1}$ and forwarded to the next block.

Lastly, the final layer of the generator $g_{L+1}$ reshapes the multivariate sequence $\tilde{Z}_L$ to a univariate time series $\hat{Z}_{i,t,\tau}$ of length $\tau = 2^{L+3}$ using a one dimensional convolution and spectral normalisation.

DISCRIMINATOR    The architecture of the discriminator mirrors the architecture of the generator. It maps the generator's output $\hat{Z}_{i,t,\tau}$ and the time features $\mathbf{X}_{t:t+\tau-1}$ to a score $d$. The first module of the discriminator $d_{L+1}$ uses a one dimensional convolution $c_1$ and a LeakyReLU activation function:

$$d'_{L+1} : \mathbb{R}^{1+D,\tau} \rightarrow \mathbb{R}^{n_f,\tau}$$
$$(\tilde{Z}_{L+1}, \mathbf{X}_{t:t+\tau-1}) \mapsto \tilde{Y}_L = \mathrm{SN}(\mathrm{LR}(c_1(\tilde{Z}_{L+1}, \mathbf{X}_{t:t+\tau-1}))) \tag{4}$$

For $i \in [L+1, 2]$, the module $d_i$ applies a downscale operator and the main function $m$:

$$d_i : \mathbb{R}^{n_f \times 2^{i+2}} \rightarrow \mathbb{R}^{n_f \times 2^{i+1}}$$
$$Y_i \mapsto Y_{i-1} = \mathrm{DOWN}(m(Y_i)) \tag{5}$$

The last module $d_1$ turns its input sequence into a score:

$$d_1 : \mathbb{R}^{n_f \times 2^3} \rightarrow \mathbb{R}$$
$$Y_1 \mapsto Y_0 = \mathrm{SN}(\mathrm{FC}(\mathrm{LR}(\mathrm{SN}(c(m(Y_1)))))) \tag{6}$$

where FC is a fully connected layer.

**PSA-GAN-C** We introduce another instantiation of `PSA-GAN` in which we forward to each generator block $g_i$ knowledge about the past. The knowledge here is a sub-series $\hat{Z}_{i,t-L_C,L_C}$ in the range $[t - L_C, t - 1]$, with $L_C$ being the context length. The context $\hat{Z}_{i,t-L_C,L_C}$ is concatenated along the feature dimension, i.e. at each time step, to the output sequence of $g_i$. It is then passed through a two layers perceptron to reshape the feature dimension and then added back to the output of $g_i$.

### 3.1    LOSS FUNCTIONS

`PSA-GAN` is trained via the LSGAN loss (Mao et al., 2017), since it has been shown to address mode collapse (Mao et al., 2017). Furthermore, least-squares type losses in embedding space have been shown to be effective in the time series domain (Mogren, 2016; Yoon et al., 2019). Additionally, we use an auxiliary moment loss to match the first and second order moments between the batch of synthetic samples and a batch of real samples:

$$\mathrm{ML}\left(\hat{\mathbf{Z}}_\tau, \mathbf{Z}_\tau\right) = \left|\mu\left(\hat{\mathbf{Z}}_\tau\right) - \mu\left(\mathbf{Z}_\tau\right)\right| + \left|\sigma\left(\hat{\mathbf{Z}}_\tau\right) - \sigma\left(\mathbf{Z}_\tau\right)\right| \tag{7}$$

where $\mu$ is the mean operator and $\sigma$ is the standard deviation operator. The real and synthetic batches have their time index and time series index aligned. We found this combination to work well for `PSA-GAN` empirically. Note that the choice of loss function was not the main focus of this study and we think that our choice can be improved in future research.

**Training procedures**   GANs are notoriously difficult to train, have hard-to-interpret learning curves, and are susceptible to mode collapse. The training procedure to address those issues together with other training and tuning details is presented in Appendix B–E.

## 4   EXPERIMENTS

The evaluation of synthetic time series data from GAN models is challenging and there is no widely accepted evaluation scheme in the time series community. We evaluate the GAN models by two guiding principles: i) Measuring to what degree the time series recover the statistics of the training dataset. ii) Measuring the performances of the GAN models in challenging, downstream forecasting scenarios.

For i), we introduce the Context-FID (Context-Frechet Inception distance) score to measure whether the GAN models are able to recover the training set statistics. The FID score is widely used for evaluating synthetic data in computer vision (Heusel et al., 2017) and uses features from an inception network (Szegedy et al., 2016) to compute the difference between the real and synthetic sample statistics in this feature space. In our case, we are interested in how well the the synthetic time series windows "fit" into the local context of the time series. Therefore, we use the time series embeddings from Franceschi et al. (2019) to learn embeddings of time series that fit into the local context. Note that we train the embedding network for each dataset separately. This allows us to directly quantify the quality of the synthetic time series samples (see Appendix D for details).

For ii), we set out to mimic several challenging time series forecasting tasks that are common for time series forecasting practitioners. These tasks often have in common that practitioners face missing or corrupted data during training or inference. Here, we set out to use the synthetic samples to complement an established baseline model, `DeepAR`, in these forecasting tasks. These tasks are: far-forecasting and missing values during inference, missing values during training, cold starts, and data augmentation. We evaluate these tasks by the normalized root mean squared distance (NRMSE). Additionally, we evaluate `PSA-GAN` model when used as a forecasting model. Note that, where applicable, we re-train our GAN models on the modified datasets to ensure that they have the same data available as the baseline model during the downstream tasks.

We also considered `NBEATS` (Oreshkin et al., 2020a) and Temporal Fusion Transformer (`TFT`) (Lim et al., 2021) as alternative forecasting models. However, we found that `DeepAR` performed best in our experiments and therefore we report these results in the main text (see Appendix F for experiment details). Please refer to Appendix G for the evaluation of NBEATS and TFT (Tables S2–S6 and Figures S3–S6).

In addition, we perform an ablation study of our model and discuss whether the Context-FID scores are indicative for downstream forecasting tasks.

### 4.1   DATASETS AND BASELINES

We use the following public, standard benchmark datasets in the time series domain: M4, hourly time series competition data (414 time series) (Makridakis et al., 2020); Solar, hourly solar energy collection data in Alabama State (137 stations) (Lai et al., 2018); Electricity, hourly electricity consumption data (370 customers) (Dheeru & Karra Taniskidou, 2017); Traffic: hourly occupancy rate of lanes in San Francisco (963 lanes) (Dheeru & Karra Taniskidou, 2017). Unless stated otherwise, we split all data into a training/test set with a fixed date and use all data before that date for training. For testing, we use a rolling window evaluation with a window size of 32 and seven windows. We minmax scale each dataset to be within $[0, 1]$ for all experiments in this paper (we scale the data back before evaluating the forecasting experiments). In lieu of finding public datasets that represent the downstream forecasting tasks, we modify each the datasets above to mimic each tasks for the respective experiment (see later sections for more details).

We compare `PSA-GAN` with different GAN models from the literature (`TIMEGAN` (Yoon et al., 2019) and `EBGAN` (Zhao et al., 2017)). In what follows `PSA-GAN-C` and `PSA-GAN` denote our proposed model with and without context, respectively. In the forecasting experiments, we use the GluonTS (Alexandrov et al., 2020) implementation of `DeepAR` which is a well-performing forecasting model and established baseline (Salinas et al., 2020).

| Dataset | Length | EBGAN | TIMEGAN | PSA-GAN | PSA-GAN-C |
|---------|--------|-------|---------|---------|-----------|
| Electricity | 64 | $2.07_{\pm 0.3}$ | $0.77_{\pm 0.12}$ | $0.018_{\pm 0.009}$ | $\mathbf{0.011}_{\pm 0.002}$ |
|  | 128 | $1.93_{\pm 0.28}$ | $1.03_{\pm 0.13}$ | $0.03_{\pm 0.01}$ | $\mathbf{0.022}_{\pm 0.01}$ |
|  | 256 | $3.02_{\pm 0.25}$ | $1.58_{\pm 0.08}$ | $0.042_{\pm 0.018}$ | $\mathbf{0.036}_{\pm 0.02}$ |
| M4 | 64 | $0.93_{\pm 0.15}$ | $0.93_{\pm 0.1}$ | $\mathbf{0.113}_{\pm 0.051}$ | $0.158_{\pm 0.088}$ |
|  | 128 | $2.53_{\pm 0.27}$ | $0.6_{\pm 0.07}$ | $\mathbf{0.156}_{\pm 0.072}$ | $0.22_{\pm 0.11}$ |
|  | 256 | $2.08_{\pm 0.13}$ | $0.89_{\pm 0.13}$ | $0.356_{\pm 0.125}$ | $\mathbf{0.235}_{\pm 0.066}$ |
| Solar-Energy | 64 | $0.48_{\pm 0.11}$ | $0.13_{\pm 0.02}$ | $0.012_{\pm 0.003}$ | $\mathbf{0.004}_{\pm 0.001}$ |
|  | 128 | $1.87_{\pm 0.43}$ | $0.22_{\pm 0.05}$ | $0.007_{\pm 0.001}$ | $\mathbf{0.002}_{\pm 0.001}$ |
|  | 256 | $2.38_{\pm 0.43}$ | $0.09_{\pm 0.02}$ | $0.064_{\pm 0.002}$ | $\mathbf{0.004}_{\pm 0.001}$ |
| Traffic | 64 | $2.65_{\pm 0.15}$ | $0.59_{\pm 0.06}$ | $0.241_{\pm 0.041}$ | $\mathbf{0.058}_{\pm 0.009}$ |
|  | 128 | $2.09_{\pm 0.26}$ | $2.13_{\pm 0.12}$ | $0.182_{\pm 0.029}$ | $\mathbf{0.044}_{\pm 0.005}$ |
|  | 256 | $1.86_{\pm 0.14}$ | $1.82_{\pm 0.12}$ | $\mathbf{0.13}_{\pm 0.017}$ | $0.488_{\pm 0.027}$ |

Table 1: Context FID-scores (lower is better) of `PSA-GAN` and baselines. We score 5120 randomly selected windows and report the mean and standard deviation.

## 4.2 DIRECT EVALUATION WITH CONTEXT-FID SCORES

Table 1 shows the Context-FID scores for `PSA-GAN`, `PSA-GAN-C` and baselines. For all sequence lengths, we find that `PSA-GAN` or `PSA-GAN-C` consistently produce the lowest Context-FID scores. For a sequence length of 256 time steps, `TIMEGAN` is the second best performing model for all datasets. Note that even though using a context in the `PSA-GAN` model results in the best overall performance, we are interested to use the GAN in downstream tasks where the context is not available. Thus, the next section will use `PSA-GAN` without context, unless otherwise stated.

## 4.3 EVALUATION ON FORECASTING TASKS

In this section, we present the results of the forecasting tasks. We find that synthetic samples do not improve over baselines in all cases. However, we view these results as a first attempt to use GAN models in these forecasting tasks and believe that future research could improve over our results.

**Far-forecasting:** In this experiment, we forecast far into the future by assuming that the data points between the training end time and the rolling evaluation window are not observed. For example, the last evaluation window would have $32 * 6$ unobserved values between the training end time and the forecast start date. This setup reflects two possible use cases: Forecasting far into the future (where no context data is available) and imputing missing data during inference because of a data outage just before forecasting. Neural network-based forecasting models such as `DeepAR` struggle under these conditions because they rely on the recent context and need to impute these values during inference. Furthermore, `DeepAR` only imputes values in the immediate context and not for the lagged values. Here, we use the GAN models during inference to fill the missing observations with synthetic data. As a baseline, we use `DeepAR` and impute the missing observations of lagged values with a moving average (window size 10) during inference. Here, we find that using the synthetic data from GAN models drastically improve over the `DeepAR` baseline and using samples from `PSA-GAN` results into the lowest NRMSE for three out of four datasets (see left Table in Figure 2). Figure 2 also shows the NRMSE as a function of the forecast window. Figure 3 shows an example of a imputed time series and the resulting forecast from the Electricity dataset when using `PSA-GAN`.

**Missing Value Stretches:** Missing values are present in many real world time series applications and are often caused by outages of a sensor or service (Zhang et al., 2019b). Therefore, missing values in real-world scenarios are often not evenly distributed over the time series but instead form missing value "stretches". In this experiment, we simulate missing value stretches and remove time series values of length 50 and 110 from the datasets. This results into 5.4-7.7% missing values for a stretch length of 50 and and 9.9-16.9% missing values for a stretch length of 110 (depending on the dataset.) Here, we split the training dataset into two parts along the time axis and only introduce missing values in the second part. We use the first (unchanged) part of the training dataset to train the GAN models and both parts of the training set to train `DeepAR`. We then use the GAN models

| NRMSE | | | | |
|---|---|---|---|---|
| Dataset
Model | Electricity | M4 | Solar | Traffic |
| DeepAR | $3.31_{\pm 0.78}$ | $0.84_{\pm 0.44}$ | $1.43_{\pm 0.02}$ | $1.0_{\pm 0.15}$ |
| EBGAN | $2.46_{\pm 0.3}$ | $1.26_{\pm 0.29}$ | $2.02_{\pm 0.07}$ | $0.9_{\pm 0.22}$ |
| TIMEGAN | $1.57_{\pm 0.25}$ | $1.32_{\pm 0.13}$ | $\mathbf{1.21}_{\pm 0.05}$ | $0.86_{\pm 0.17}$ |
| PSA-GAN | $\mathbf{0.99}_{\pm 0.44}$ | $\mathbf{0.62}_{\pm 0.3}$ | $1.22_{\pm 0.05}$ | $\mathbf{0.59}_{\pm 0.2}$ |

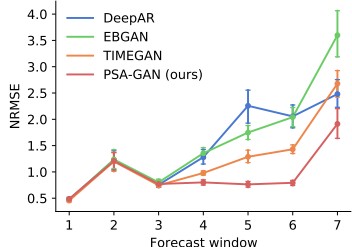

Figure 2: NRMSE of far-forecasting experiments (lower is better). Mean and confidence intervals are obtained by re-running each method ten times. **Left**: NRMSE average over different forecast windows. `DeepAR` is only run on the real time series data. The other models correspond to `DeepAR` and one of the GAN models for filling missing observations. **Right**: NRMSE by forecast window. The first window will have no missing values during inference (we forecast the first 32 steps after the training range) and we increase the missing values during inference with each window (the last window will have $32 * 6$ missing values). For `DeepAR`, the NRMSE increases noticeably at the fourth forecast window while using `PSA-GAN` to impute the missing values at inference keeps the NRMSE low.

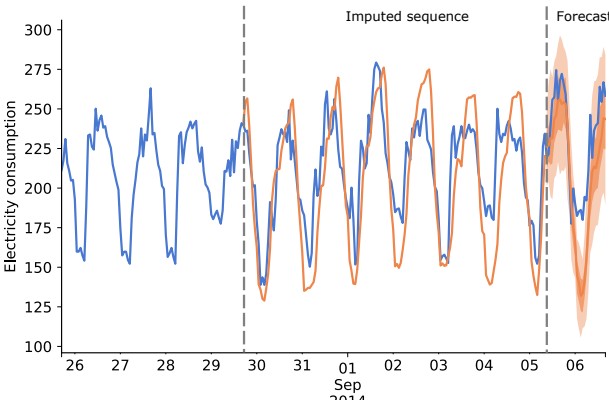

Figure 3: Example of the far-forecasting experiment for the Electricity dataset. True data is shown in blue and the imputed sequence and forecasts of the model in orange. The orange line between the dashed lines corresponds to the `PSA-GAN` imputed sequence and is used by `DeepAR` for forecasting. Note that this part is unobserved in this experiment. The imputed sequence allows to generate a reasonable forecast, even if many data points before the forecast start date are unobserved.

to impute missing values during training and inference of `DeepAR`. Figure 4 shows that using the GAN models to impute missing values during *training* `DeepAR` on data with missing value stretches reduces its forecasting error. While all GAN models reduce the NRMSE in this setup, `PSA-GAN` is most effective in reducing the error in this experiment. See Figure S1 for a detailed split by dataset.

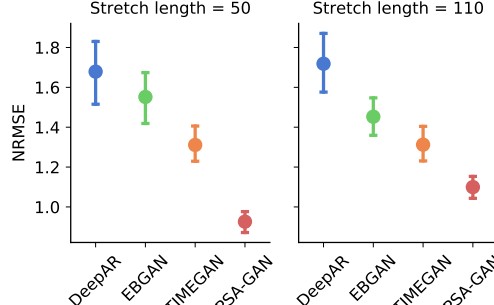

Figure 4: Performance of using different GAN models for imputation in the missing value stretch length experiment. Markers denote the mean NRMSE averaged over datasets and error bars the 68% confidence interval over ten runs. For 50 and 110 missing value stretch length, using `PSA-GAN` to impute missing values results in the lowest overall error, whereas `DeepAR` returns the highest.

**Cold Starts:** In this experiment, we explore whether the GAN models can support `DeepAR` in a cold start forecasting setting. Cold starts refer to new time series (like a new product in demand forecasting) for which little or no data is available. Here, we randomly truncate 10%, 20%, and 30% of the time series from our datasets such that only the last 24 (representing 24 hours of data) values before inference are present. We only consider a single forecasting window in this experiment. We then again use the GAN models to impute values for the lags and context that `DeepAR` conditions for

| NRMSE | | | | |
|---|---|---|---|---|
| Dataset Model | Electricity | M4 | Solar | Traffic |
| DeepAR | **0.49**±0.03 | **0.26**±0.02 | 1.07±0.01 | **0.45**±0.01 |
| EBGAN | 0.64±0.05 | 0.32±0.09 | 1.18±0.03 | 0.50±0.03 |
| TIMEGAN | 0.56±0.05 | 0.28±0.02 | **1.06**±0.02 | 0.47±0.004 |
| PSA-GAN | 0.64±0.22 | 0.3±0.04 | 1.07±0.03 | 0.50±0.11 |

Table 2: NRMSE accuracy comparison of data augmentation experiments (lower is better, best method in bold). Mean and 95% confidence intervals are obtained by re-running each method five times. DeepAR is only run on the real time series data. The other models correspond to DeepAR and one of the GAN models for data augmentation.

forecasting the cold start time series. Figure 5 shows the forecasting error of the different models for the cold start time series only. In this experiment, PSA-GAN and TIMEGAN improve the NRMSE over DeepAR and are on-par overall (mean NMRSE 0.70 and 0.71 for PSA-GAN and TIMEGAN, respectively). See Figure S2 for a detailed split by dataset.

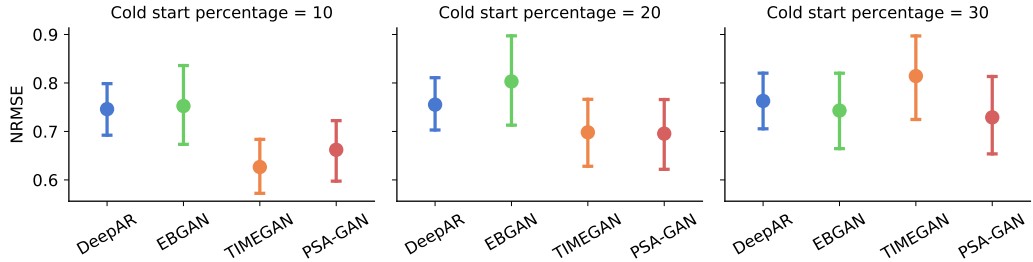

Figure 5: Performance of using different GAN models for imputation in the cold start experiment. Markers denote the mean NRMSE averaged over datasets and error bars the 68% confidence interval over ten runs. Note that this figure only shows the error of the cold start time series. Overall, TIMEGAN and PSA-GAN improve the NRMSE over DeepAR in this setup, while PSA-GAN is the best method when 30% of time series are cold starts.

**Data Augmentation:** In this experiment, we average the real data and GAN samples to augment the data during training. [1] During inference, DeepAR is conditioned on real data only to generate forecasts. In Table 2 we we can see that none of the GAN models for data augmentation consistently improves over DeepAR. Overall, TIMEGAN is the best-performing GAN model but plain DeepAR still performs better. This finding is aligned with recent work in the image domain where data augmentation with GAN samples does not improve a downstream task (Ravuri & Vinyals, 2019). We hypothesize that the GAN models cannot improve in the data augmentation setup because they are trained to generate realistic samples and not necessarily to produce relevant invariances. Furthermore, the dataset sizes might be sufficient for DeepAR to train a well-performing model and therefore the augmentation might not be able to reduce the error further. More research is required to understand whether synthetic data can improve forecasting models via data augmentation.

**Forecasting Experiments:** In this experiment, we use the GAN models directly for forecasting (Table 3, example samples in Appendix H). We can see that DeepAR consistently performs best. This is expected as DeepAR takes into account context information and lagged values. This kind of information is not available to the GAN models. To test this we further consider PSA-GAN-C, i.e. PSA-GAN with 64 previous time series values as context, and further evaluate DeepAR with drawing lags only from the last 64 values (DeepAR-C). We can see that in this case PSA-GAN-C outperforms DeepAR-C in 3 out of 4 datasets and PSA-GAN performs on par with DeepAR-C. Moreover, both PSA-GAN and PSA-GAN-C are the best performing GAN models. Adding lagged values to the GAN models as context could further improve their performance and adversarial/attention architectures have been previously used for forecasting (Wu et al., 2020).

---

[1]We also tried using only the GAN generated data for training and experimented with ratios of mixing synthetic and real samples, similar to the work in (Ravuri & Vinyals, 2019). Furthermore, we tried different weights and scheduled sampling but this did not improve the results.

| | NRMSE | | | |
|---|---|---|---|---|
| Dataset
Model | Electricity | M4 | Solar | Traffic |
| DeepAR | $\mathbf{0.49}_{\pm 0.03}$ | $\mathbf{0.26}_{\pm 0.02}$ | $\mathbf{1.07}_{\pm 0.01}$ | $\mathbf{0.45}_{\pm 0.01}$ |
| DeepAR-C | $1.43_{\pm 0.40}$ | $0.73_{\pm 0.46}$ | $0.95_{\pm 0.01}$ | $0.89_{\pm 0.16}$ |
| EBGAN | $3.63_{\pm 0.07}$ | $1.14_{\pm 0.01}$ | $2.31_{\pm 0.01}$ | $2.02_{\pm 0.00}$ |
| TIMEGAN | $3.04_{\pm 0.00}$ | $1.12_{\pm 0.00}$ | $1.83_{\pm 0.00}$ | $1.13_{\pm 0.00}$ |
| PSA-GAN | $1.53_{\pm 0.00}$ | $0.61_{\pm 0.00}$ | $1.35_{\pm 0.00}$ | $0.66_{\pm 0.00}$ |
| PSA-GAN-C | $1.18_{\pm 0.00}$ | $0.33_{\pm 0.00}$ | $1.32_{\pm 0.00}$ | $0.57_{\pm 0.00}$ |

Table 3: NRMSE accuracy comparison of forecasting experiments (lower is better, best method in bold). Mean and 95% confidence intervals are obtained by re-running each method five times. DeepAR-C and PSA-GAN-C use the same 64 previous values as context. Among GAN models PSA-GAN and PSA-GAN-C perform best.

## 4.4 ABLATION STUDY

Figure 6 shows the results of our ablation study where we disable important components of our model: moment loss, self-attention, and fading in of new layers. We measure the performance of the ablation models by Context-FID score. Overall, our propost logog ed PSA-GAN model performs better than the ablations which confirms that these components contribute to the performance of the model.

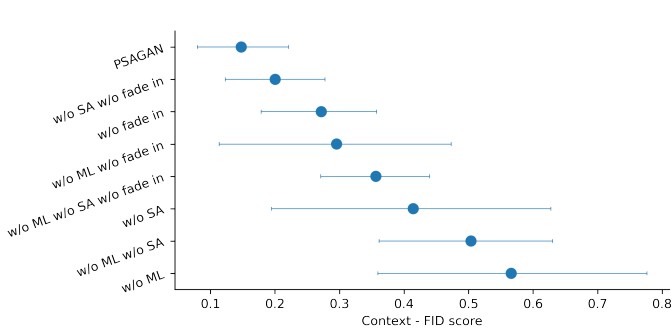

Figure 6: We perform an ablation study by disabling moment loss, self-attention, and fading in of new layers of PSA-GAN. We show the mean performance over three runs on four datasets and the 68% confidence interval. PSA-GAN has the lowest Context-FID score which confirms that the proposed model requires these components for good performance. **SA**: Self-attention, **ML**: Moment loss, **fade in**: Progressive fade in of new layers

## 4.5 LOW CONTEXT-FID SCORE MODELS CORRESPOND TO BEST-PERFORMING FORECASTING MODELS

One other observation is that the lowest Context-FID score models correspond to the best models in the data augmentation and far-forecasting experiments. PSA-GAN and TIMEGAN produce the lowest Context-FID samples and both models also improve over the baseline in most downstream tasks. Overall, PSA-GAN has lowest Context-FID and also outperforms the other models in the downstream forecasting tasks, except for the cold start task. Additionally, we calculated the Context-FID scores for the ablation models mentioned in Figure 6 (but with a target length of 32) and the NRMSE of these models in the forecasting experiment (as in Table 3). We find that the Pearson correlation coefficient between the Context-FID and forecasting NRMSE is 0.71 and the Spearman's rank correlation coefficient is 0.67, averaged over all datasets. All datasets (except Traffic) have a correlation coefficient of at least 0.73 in either measure (see Table S1 in the Appendix).

## 5 CONCLUSION

We have presented PSA-GAN, a progressive growing time series GAN augmented with self-attention, that produces long realistic time series and improves downstream forecasting tasks that are challenging for deep learning-based time series models. Furthermore, we introduced the Context-FID score to assess the quality of synthetic time series samples produced by GAN models. We found that the lowest Context-FID scoring models correspond to the best-performing models in downstream tasks. We believe that time series GANs that scale to long sequences combined with a reliable metric to assess their performance might lead to their routine use in time series modeling.

## 6 REPRODUCIBILITY STATEMENT

Details necessary for reproducing our experiments are given in the Appendix. In particular, details on training are provided in Sections B and C, together with hyperparameter tuning in Section E and further experimental settings in Section F. The code we used in the paper is available under: `https://github.com/mbohlkeschneider/psa-gan` and we will additionally disseminate `PSA-GAN` via GluonTS: `https://github.com/awslabs/gluon-ts`.

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

APPENDIX

## A    TIME SERIES FEATURES

We use time-varying features to represent the time dimension for the GAN models to enable training on time series windows sampled from the full-length time series . We use `HourOfDay`, `DayOfWeek`, `DayOfMonth` and `DayOfYear` features. Each feature is encoded using a single index number (for example, between $[0, 365[$ for `DayOfYear`) and normalized to be within [-0.5, 0.5]. We also use an age feature to represent the age of the time series to be $log(2.0 + t)$ where $t$ is the time index. Additionally, we embed the index of the individual time series into a 10 dimensional vector with an embedding network to and use it as a non time-varying feature. `DeepAR` uses the same features in forecasting.

## B    TRAINING PROCEDURE

`PSA-GAN` uses spectral normalization and progressive fade-in of new layers to stabilize training.

**Progressive fade in of new layers**    As training progresses, new layers are added to double the length of time series. However, simply adding new untrained layers drastically changes the number of parameters and the loss landscape, which destabilizes training. Progressive fade in of new layers is a technique introduce by Karras et al., 2017 to mitigate this effect where each layer $g_i$ and $d_i$ is smoothly introduced. For $i \in [2, L]$ the fade in of new layers is written as:

$$
\begin{aligned}
g_i :&\mathbb{R}^{n_f \times 2^{i+1}} \to \mathbb{R}^{n_f \times 2^{i+2}} \\
&\tilde{Z}_{i-1} \mapsto \tilde{Z}_i = \alpha m(\text{UP}(\tilde{Z}_{i-1})) + (1 - \alpha)\,\text{UP}(\tilde{Z}_{i-1})
\end{aligned}
\tag{8}
$$

$$
\begin{aligned}
d_i :&\mathbb{R}^{n_f \times 2^{i+2}} \to \mathbb{R}^{n_f \times 2^{i+1}} \\
&Y_i \mapsto Y_{i-1} = \alpha\,\text{DOWN}(m(Y_i)) + (1 - \alpha)\,\text{DOWN}(Y_i)
\end{aligned}
\tag{9}
$$

where $\alpha$ is a scalar initialized to be zero and grows linearly to one over 500 epochs.

**Spectral Normalisation**    Both the generator and the discriminator benefit from using spectral normalisation layers. In the discriminator, Spectral Normalisation stabilizes training (Miyato et al., 2018). It constrains the Lipschitz constant of the discriminator to be bounded by one. In the generator the spectral normalisation stabilises the training and avoids escalation of the gradient magnitude (Zhang et al., 2019a).

## C    TRAINING DETAILS

All GAN models in this paper have been implemented using PyTorch from Paszke et al. (2019).

PSA-GAN has been trained over 6500 epochs, where a new block is added every 1000 epochs and is faded over 500 epochs. At each epoch, PSA-GAN trains on 100 batches of size 512. It optimises its parameters using the Adam (Kingma & Ba, 2014) with a learning rate of 0.0005 and betas of $(0.9, 0.999)$ for both the generator and the discriminator.

PSA-GAN (and its variants) has been trained on ml.p2.xlarge Amazon instances for 21 hours.

The number of in-channels and out-channels $n_f$ in the Main Block (Figure 1, *Right*) equals 32. The number of out-channels in the convolution layer $c$ equals 1 (Eq.6 ). The number of time features $D$ equals 15.

For the case of PSA-GAN-C, the MLP block is a two layers perceptron with LeakyReLU. It incorporates knowledge of the past by selecting the $L_C = 64$ time points and concatenates them with its input.

## D    CONTEXT-FID SCORE ON TIME SERIES EMBEDDINGS

To compute the Context-FID, we replace InceptionV3 (Szegedy et al., 2016) with the encoder $E$ of Franceschi et al. (2019), which we train separately for each dataset.

Our proposed Context-FID score is computed as follows: we first select a time range $[t, t + \tau]$. We then sample a batch of synthetic time series $\hat{\mathbf{Z}}_{t,\tau} = [\hat{Z}_{1,t,\tau}, ..., \hat{Z}_{N,t,\tau}]$ and a batch of real time series $\mathbf{Z}_{t,\tau} = [Z_{1,t,\tau}, ..., Z_{N,t,\tau}]$ that we encode with $E$ into $\hat{\mathbf{Z}}^e_{t,\tau}$ and $\mathbf{Z}^e_{t,\tau}$ respectively. Finally, we compute the FID score of the embeddings.

## E    HYPERPARAMETER TUNING

We did limited hyperparameter tuning in this study to find default hyperparemters that perform well across datasets. We performed the following grid-search: the `batch size`, the `number of batches used per epoch`, the presence of `Self-attention` and the presence of the `Progressive fade in of new layers` mechanism explained in section B, paragraph B. The range considered for each hyper-parameter is: `batch size`: $[64, 128, 256, 512]$, the `number of batches used per epoch`: $[50, 100, 200]$, the presence of `Self-attention`: [True, False] and the presence of the `Progressive fade in of new layers`: [True, False].

The set of hyperparameters is shared across the four datasets we train PSA-GAN on (which means we do not tune hyperparameters per datasets). We select the final hyperparameter set by lowest Context-FID score on the training set, averaged across all datasets.

## F    FORECASTING EXPERIMENT DETAILS

We use the `DeepAR` (Salinas et al., 2020) implementation in `GluonTS` (Alexandrov et al., 2020) for the forecasting experiments. We use the default hyperparameters of `DeepAR` with the following exceptions: `epochs=100, num batches per epoch=100, dropout rate=0.01, scaling=False, prediction length=32, context length=64, use feat static cat=True`. We use the Adam optimizer with a learning rate of $1\mathrm{e}{-}3$ and weight decay of $1\mathrm{e}{-}8$ . Additionally, we clip the gradient to 10.0 and reduce the learning rate by a factor of 2 for every 10 consecutive updates without improvement, up to a minimum learning rate of $5\mathrm{e}{-}5$. `DeepAR` also uses lagged values from previous time steps that are used as input to the LSTM at each time step $t$. We set the time series window size that `DeepAR` samples to 256 and truncate the default hourly lags to fit the context window, the prediction window, and the longest lag into the window size 256. We use the following lags: [1, 2, 3, 4, 5, 6, 7, 23, 24, 25, 47, 48, 49, 71, 72, 73, 95, 96, 97, 119, 120, 121, 143, 144, 145].

For NBEATS (Oreshkin et al., 2020a), we use the default parameters and set the trainer settings to `epochs=100, num batches per epoch=50`. Note that in the original paper, NBEATS is an ensemble of 180 networks. Due to the high computational cost, we only ensemble three networks with the settings `context length=64, context length=96, context length=128` and mean absolute percentage error as the loss function.

For TFT (Lim et al., 2021), we use the default hyperparameters except: `epochs=100, num batches per epoch=50, context length=128`.

For all methods, we used the implementations available in GluonTS (Alexandrov et al., 2020).

## G    ADDITIONAL EXPERIMENTS

|  | Pearson | Spearman |
|---|---|---|
| Traffic | 0.597 | 0.609 |
| M4 | 0.858 | 0.455 |
| Solar | 0.852 | 0.773 |
| Electricity | 0.533 | 0.891 |

Table S1: Pearson and Spearman rank-order correlation coefficient between the Context-FID score and NRMSE for different datasets. These results suggest that low Context-FID scores indicate low error in forecasting and therefore Context-FID could be a useful score to develop GAN models in the time series domain.

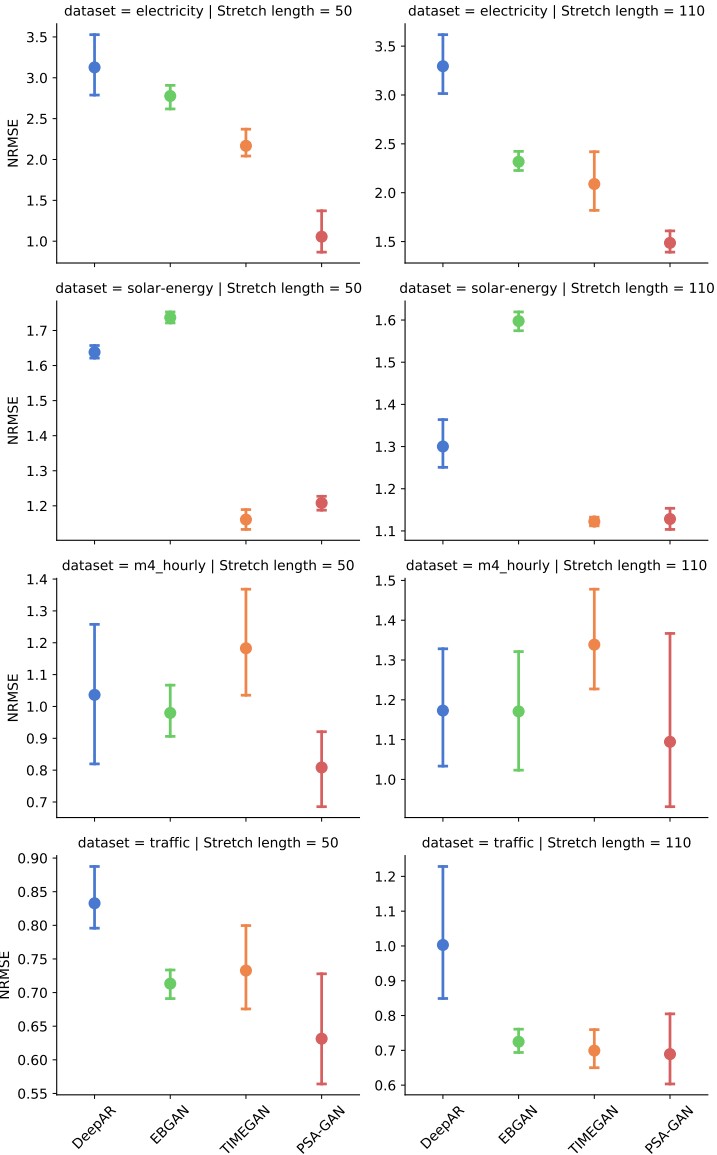

Figure S1: Detailed performance of using different GAN models for imputation in the missing value stretch length experiment. Markers denote the mean NRMSE averaged over ten runs and error bars represent the 95% confidence interval. For 50 and 110 missing value stretch length, using `PSA-GAN` to impute missing values results in the lowest error for three out of four datasets.

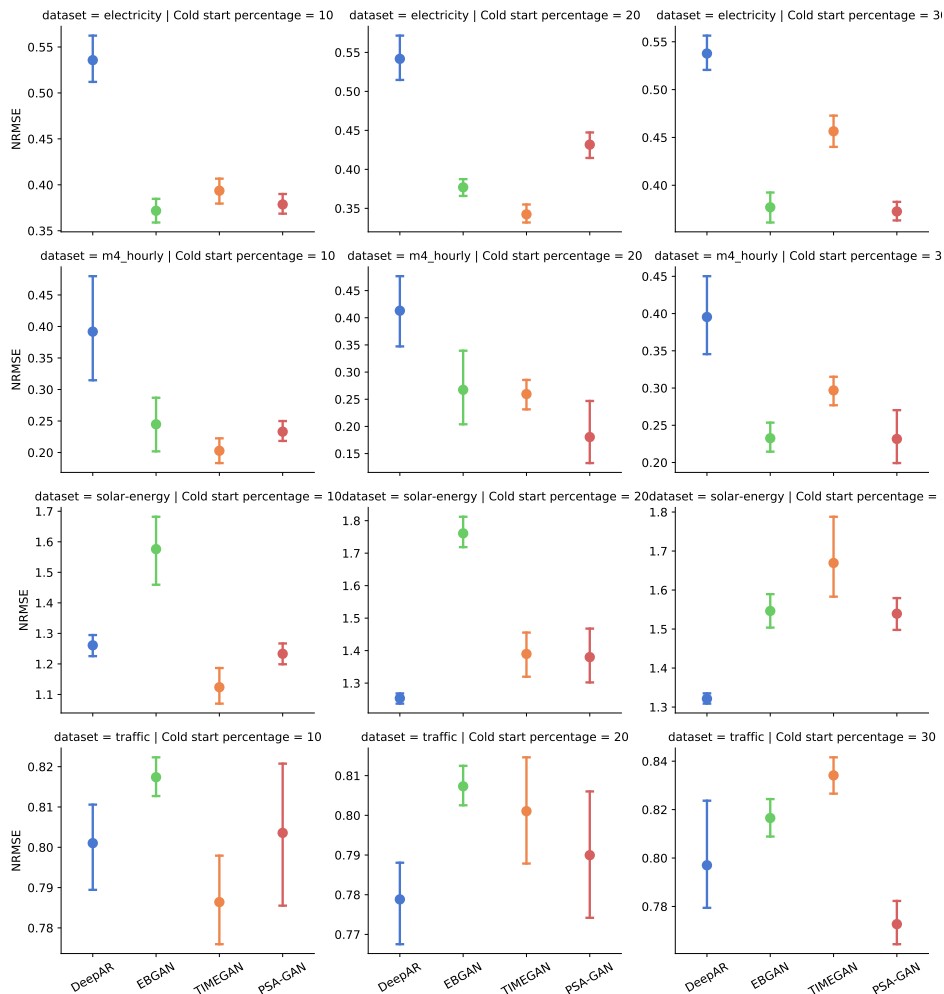

Figure S2: Detailed performance of using different GAN models for the cold start experiment. Markers denote the mean NRMSE averaged over ten runs and error bars the 95% confidence interval. `TIMEGAN` and `PSA-GAN` improve over `DeepAR` in three out of four datasets.

| Dataset Model | NRMSE Electricity | M4 | Solar | Traffic |
|---|---|---|---|---|
| NBEATS | $5.29 \pm 0.28$ | $1.92 \pm 0.04$ | $2.08 \pm 0.0$ | $0.94 \pm 0.02$ |
| EBGAN | $3.59 \pm 0.21$ | $1.76 \pm 0.03$ | $2.07 \pm 0.0$ | $0.96 \pm 0.0$ |
| TIMEGAN | $2.06 \pm 0.02$ | $1.75 \pm 0.01$ | $2.07 \pm 0.0$ | $0.8 \pm 0.01$ |
| PSA-GAN | $\mathbf{1.30} \pm 0.04$ | $\mathbf{0.4} \pm 0.01$ | $2.07 \pm 0.0$ | $\mathbf{0.65} \pm 0.0$ |

Table S2: NRMSE of far-forecasting experiments with NBEATS as the forecasting model (lower is better). Mean and confidence intervals are obtained by re-running each method ten times.

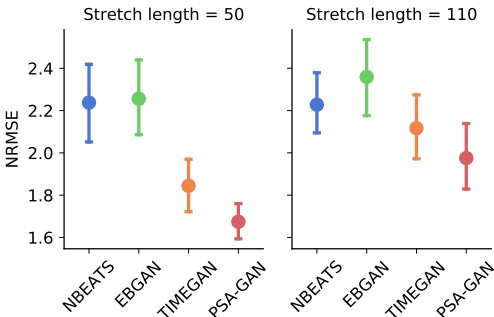

Figure S3: Performance of using different GAN models for imputation in the missing value stretch length experiment using NBEATS as the forecasting model. Markers denote the mean NRMSE averaged over datasets and error bars the 68% confidence interval over ten runs.

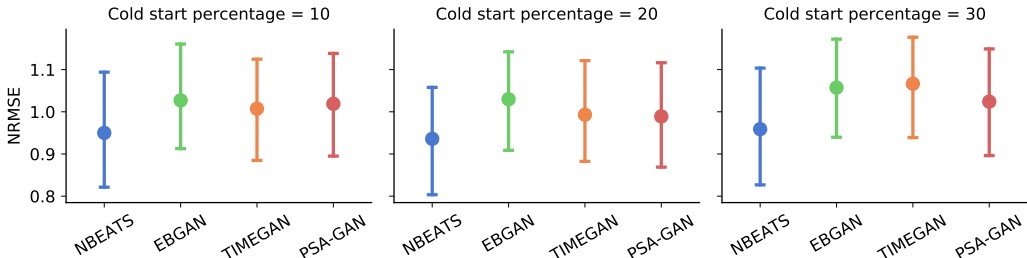

Figure S4: Performance of using different GAN models for imputation in the cold start experiment using NBEATS as the forecasting model. Markers denote the mean NRMSE averaged over datasets and error bars the 68% confidence interval over ten runs. Note that this figure only shows the error of the cold start time series.

| Dataset Model | NRMSE Electricty | M4 | Solar | Traffic |
|---|---|---|---|---|
| NBEATS | $0.86 \pm 0.06$ | $\mathbf{0.28} \pm 0.01$ | $2.07 \pm 0.0$ | $0.58 \pm 0.0$ |
| EBGAN | $0.74 \pm 0.03$ | $0.31 \pm 0.01$ | $2.08 \pm 0.0$ | $0.58 \pm 0.01$ |
| TIMEGAN | $\mathbf{0.58} \pm 0.01$ | $0.28 \pm 0.01$ | $2.08 \pm 0.0$ | $\mathbf{0.51} \pm 0.0$ |
| PSA-GAN | $0.92 \pm 0.06$ | $0.3 \pm 0.01$ | $\mathbf{1.97} \pm 0.02$ | $0.54 \pm 0.01$ |

Table S3: NRMSE accuracy comparison of data augmentation experiments (lower is better) using NBEATS as the forecasting model. Mean and 95% confidence intervals are obtained by re-running each method five times.

| Dataset Model | NRMSE Electricity | M4 | Solar | Traffic |
|---|---|---|---|---|
| TFT | $4.51 \pm 0.02$ | $1.78 \pm 0.04$ | $1.74 \pm 0.02$ | $0.94 \pm 0.0$ |
| EBGAN | $3.54 \pm 0.22$ | $1.5 \pm 0.05$ | $2.5 \pm 0.13$ | $0.76 \pm 0.02$ |
| TIMEGAN | $2.03 \pm 0.05$ | $1.71 \pm 0.04$ | $\mathbf{1.26} \pm 0.03$ | $0.82 \pm 0.03$ |
| PSA-GAN | $\mathbf{1.97} \pm 0.05$ | $\mathbf{0.34} \pm 0.01$ | $2.15 \pm 0.13$ | $\mathbf{0.70} \pm 0.0$ |

Table S4: NRMSE of far-forecasting experiments with Temporal Fusion Transformer (TFT) as the forecasting model (lower is better). Mean and confidence intervals are obtained by re-running each method ten times.

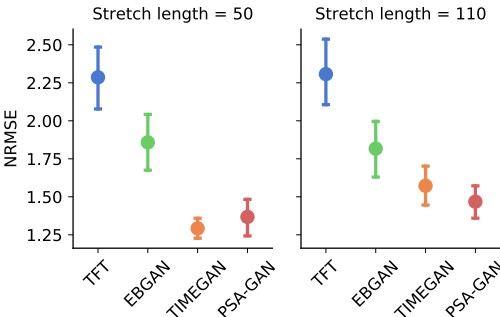

Figure S5: Performance of using different GAN models for imputation in the missing value stretch length experiment using Temporal Fusion Transformer (TFT) as the forecasting model. Markers denote the mean NRMSE averaged over datasets and error bars the 68% confidence interval over ten runs.

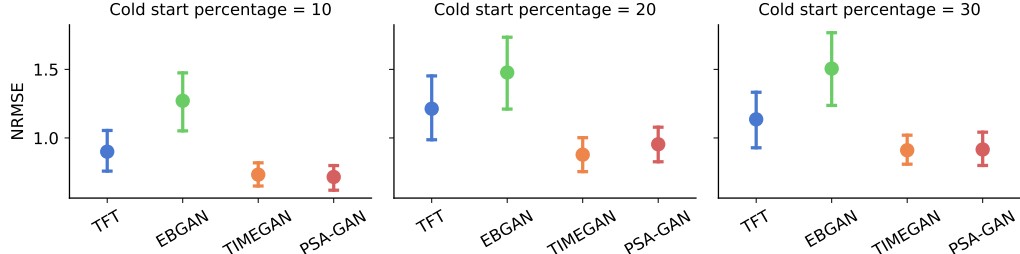

Figure S6: Performance of using different GAN models for imputation in the cold start experiment using Temporal Fusion Transformer (TFT) as the forecasting model. Markers denote the mean NRMSE averaged over datasets and error bars the 68% confidence interval over ten runs. Note that this figure only shows the error of the cold start time series.

| Dataset Model | NRMSE Electricity | M4 | Solar | Traffic |
|---|---|---|---|---|
| TFT | $\mathbf{0.63} \pm 0.04$ | $\mathbf{0.26} \pm 0.02$ | $1.29 \pm 0.04$ | $\mathbf{0.40} \pm 0.0$ |
| EBGAN | $0.9 \pm 0.05$ | $0.31 \pm 0.01$ | $\mathbf{1.21} \pm 0.05$ | $0.64 \pm 0.01$ |
| TIMEGAN | $0.83 \pm 0.04$ | $0.85 \pm 0.06$ | $1.28 \pm 0.03$ | $0.51 \pm 0.01$ |
| PSA-GAN | $0.94 \pm 0.03$ | $0.35 \pm 0.02$ | $1.64 \pm 0.04$ | $0.42 \pm 0.0$ |

Table S5: NRMSE accuracy comparison of data augmentation experiments (lower is better) using Temporal Fusion Transformer (TFT) as the forecasting model. Mean and 95% confidence intervals are obtained by re-running each method five times.

| NRMSE | | | | |
|---|---|---|---|---|
| Dataset
Model | Electricity | M4 | Solar | Traffic |
| DeepAR | $\mathbf{0.49}_{\pm 0.03}$ | $\mathbf{0.26}_{\pm 0.02}$ | $\mathbf{1.07}_{\pm 0.01}$ | $0.45_{\pm 0.01}$ |
| NBEATS | $0.86_{\pm 0.06}$ | $0.28_{\pm 0.01}$ | $2.07_{\pm 0.00}$ | $0.58_{\pm 0.00}$ |
| TFT | $0.63_{\pm 0.04}$ | $0.26_{\pm 0.02}$ | $1.29_{\pm 0.04}$ | $\mathbf{0.40}_{\pm 0.00}$ |
| DeepAR-C | $1.43_{\pm 0.40}$ | $0.73_{\pm 0.46}$ | $0.95_{\pm 0.01}$ | $0.89_{\pm 0.16}$ |
| EBGAN | $3.63_{\pm 0.07}$ | $1.14_{\pm 0.01}$ | $2.31_{\pm 0.01}$ | $2.02_{\pm 0.00}$ |
| TIMEGAN | $3.04_{\pm 0.00}$ | $1.12_{\pm 0.00}$ | $1.83_{\pm 0.00}$ | $1.13_{\pm 0.00}$ |
| PSA-GAN | $1.53_{\pm 0.00}$ | $0.61_{\pm 0.00}$ | $1.35_{\pm 0.00}$ | $0.66_{\pm 0.00}$ |
| PSA-GAN-C | $1.18_{\pm 0.00}$ | $0.33_{\pm 0.00}$ | $1.32_{\pm 0.00}$ | $0.57_{\pm 0.00}$ |

Table S6: Extended NRMSE accuracy comparison of forecasting experiments with NBEATS and Temporal Fusion Transformer (TFT) as additional forecasting models (lower is better).

## H  SAMPLE FORECASTS

We present more sample forecasts under the framework of the forecasting experiment. In Figures S7–S10 we present sample forecast for Electricity, M4, Solar Energy, and Traffic datasets, respectively. We can see that the synthetic time series generated by `PSA-GAN` and `PSA-GAN-C` consistently convey a better approximation to the ground truth values. One can further observe that this task is particularly challenging for GANs as data generation is located in a time-frame that is not covered while training.

Figure S7: Sample forecasts for dataset **Electricity**

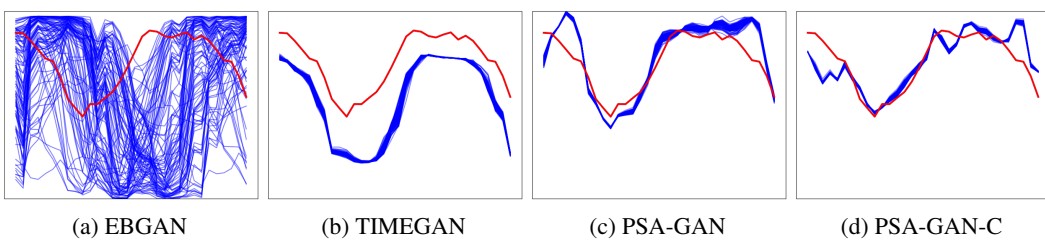

|             |             |             |             |
|:-----------:|:-----------:|:-----------:|:-----------:|
| (a) EBGAN   | (b) TIMEGAN | (c) PSA-GAN | (d) PSA-GAN-C |

Figure S8: Sample forecasts for dataset **M4**

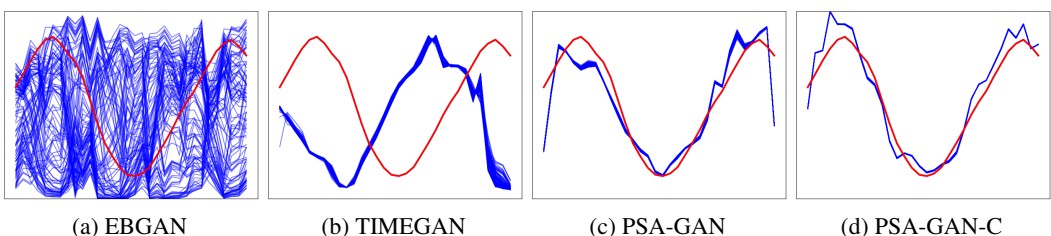

|             |             |             |             |
|:-----------:|:-----------:|:-----------:|:-----------:|
| (a) EBGAN   | (b) TIMEGAN | (c) PSA-GAN | (d) PSA-GAN-C |

Figure S9: Sample forecasts for dataset **Solar**

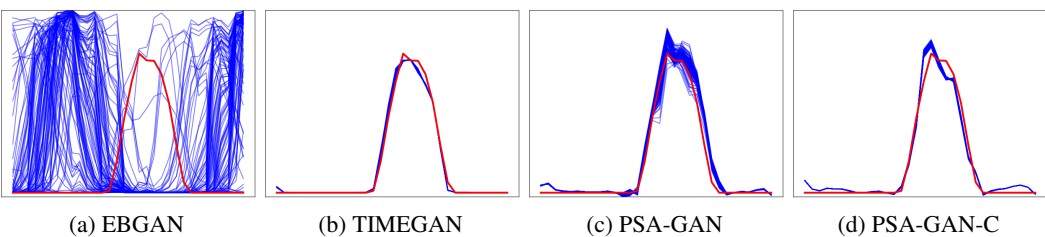

|             |             |             |             |
|:-----------:|:-----------:|:-----------:|:-----------:|
| (a) EBGAN   | (b) TIMEGAN | (c) PSA-GAN | (d) PSA-GAN-C |

Figure S10: Sample forecasts for dataset **Traffic**

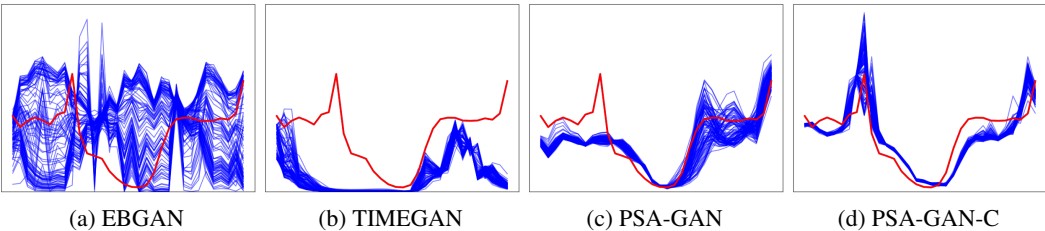

|             |             |             |             |
|:-----------:|:-----------:|:-----------:|:-----------:|
| (a) EBGAN   | (b) TIMEGAN | (c) PSA-GAN | (d) PSA-GAN-C |

