# OpenReview forum: "PSA-GAN: Progressive Self Attention GANs for Synthetic Time Series"
_ICLR.cc/2022/Conference — ICLR 2022 Poster_

### Official Review · Reviewer_4v5L · 2021-10-24

**Correctness:** 3
**Technical Novelty And Significance:** 2
**Empirical Novelty And Significance:** 2
**Recommendation:** 6
**Confidence:** 4

**Main Review:**

- The paper adapts the known concepts for image GAN literature to time-series, without substantial extra inventions. It poses a concern for novelty.

- The architecture is just composed of standard layers from image GAN literature, based on self-attention. Whereas for the experiments, the Author use the RNN-based DeepAR. What happens if an RNN based architecture is used for generation or if a self-attention based architecture is used for forecasting?

- The loss function choice seems heuristic and not very well explained. How does the loss help with periodicity? What happens if the signals have high vs. low frequency components? What happens when the signals have peaks? Different impactful time-series cases should be considered and the section should be expanded accordingly.

- I don't think the FID score would have significant value for time-series as the perceptual quality of time-series to humans is different than images. Beyond proposing the score, its usefulness should be demonstrated.

- The procedure for hyperparameter tuning is not well explained. How does one construct a validation set and what validation reward to use?

- I think experiments section should be better organized and the writing should be improved for a more clear presentation. The results on far-forecasting and data imputation seem quite promising. The improvements in cold start and data augmentation seem negligible. For forecasting, the gap with DeepAR is still significant. Overall, rather than focusing on a lot of results, I suggest emphasizing on the key capabilities the data synthesis brings, with significant improvements. Fig. 6 is just a cherry-picked example and it does not tell much.

- For data augmentation, how can one optimize what ratio of synthetic data to use and is there a way of more intelligent sampling of synthetic data specifically focusing on the low data coverage regimes? The naive application does not seem promising and the authors need to think further to show strong results.

- Many recent papers show significant outperformance over DeepAR. That aspect is orthogonal to the contributions of the paper but I suggest benchmarking with at least one more forecasting model. Specifically for far-forecasting, RNN based models are known to suffer from error propagation without teacher forcing. So direct decoding approaches that generate all the multi-horizon forecasts would be better candidates, and likely will reduce the gains with synthetic data.

**Summary Of The Paper:**

PSA-GAN is proposed, based on progressively generating the output time-series with self-attention. Via extensive experiments, its benefits in synthesizing realistic samples are demonstrated.

**Summary Of The Review:**

Overall, the paper has strong experimental results and solid contributions. But the novelty is not very high, and also several aspects of the paper need improvements. I am willing to increase my score if the authors address the points above.

---------------------------

I am updating my score as the reviewers have addressed some of my concerns with extra experiments. I believe the paper still needs significant edits to reflect the points in the reviewer responses.

---

> ### Author Response · Authors · 2021-11-18
> **Response to Reviewer 4v5L - Part 1**
>
> We thank the reviewer for the insightful comments and suggested improvements of our work.  Where possible, we performed additional experiments to address the comments. Due to the time constraints in the discussion period, we have not been able to finish all experiments but are committed to do so within the discussion period. Please find our response to the individual concerns below.
>
> > The paper adapts the known concepts for image GAN literature to time-series, without substantial extra inventions. It poses a concern for novelty.
>
> We thank the reviewer for this comment. We agree that the components of the PSA-GAN model (progressive growing, self-attention, least-squares loss, moment loss)  have been proposed and studied before. However, we believe that this combination is non-trivial and has not been proposed in the time series domain, yet. Additionally, our paper makes contributions on the non-trivial problem on how to evaluate GAN models in the time series domain. To this end, we propose the Context-FID score based on unsupervised time series embeddings. We have performed additional analysis and updated the paper to show that the Context-FID score correlates with the forecasting performance (NRMSE) of the GAN model (p.9).Additionally, we evaluate the GAN models by showing their added value in four challenging forecasting scenarios (missing values during inference/far-forecasting, missing value stretches, cold-starts, data augmentation), thus providing practitioners with a blue-print on how to leverage time series GAN models. Overall, we think that our paper makes contributions beyond the PSA-GAN model. Specifically, our contributions on GAN model evaluation and their use in forecasting are highly relevant and have not been studied extensively in the time series domain.
>
> > The architecture is just composed of standard layers from image GAN literature, based on self-attention. Whereas for the experiments, the Author use the RNN-based DeepAR. What happens if an RNN based architecture is used for generation or if a self-attention based architecture is used for forecasting?
>
> We thank the reviewer for proposing this experiment. We have implemented our setup with another attention-based forecasting model named Temporal Fusion Transformer (TFT) (https://arxiv.org/abs/1912.09363, to appear in the International Journal of Forecasting). Note that TimeGAN is an RNN-based GAN model. Thus, the combination of TimeGAN and TFT is an instantiation of an RNN GAN model and an attention forecasting model that the reviewer proposed.
> At the time of the response, we completed the experiments in the far-forecasting experiment with this setup:
>
> | model   | (NRMSE, Electricity)   | (NRMSE, M4)   | (NRMSE, Solar)   | (NRMSE, Traffic)   |
> |:--------|:---------------------------|:-------------------------|:----------------------------|:-----------------------|
> | TFT     | 4.51 +\- 0.02              | 1.78 +\- 0.04            | 1.74 +\- 0.02               | 0.94 +\- 0.0           |
> | EBGAN      | 3.54 +\- 0.22              | 1.5 +\- 0.05             | 2.5 +\- 0.13                | 0.76 +\- 0.02          |
> | TIMEGAN      | 2.03 +\- 0.05              | 1.71 +\- 0.04            | 1.26 +\- 0.03               | 0.82 +\- 0.03          |
> | PSA-GAN (ours)      | 1.97 +\- 0.05              | 0.34 +\- 0.01            | 2.15 +\- 0.13               | 0.7 +\- 0.0            |
>
> Overall, the TIMEGAN/TFT combination works well for Solar and Electricity, while the PSA-GAN/TFT combination works best for Electricity and Traffic. Interestingly, only TIMEGAN/TFT work well for Solar which suggests that for this dataset a RNN/attention of GAN/Forecasting model (or vice versa) is most effective.
>
> We aim to complete the remaining experiments for TFT by the end of the discussion period and include the results in the Appendix of the paper.

---

> ### Author Response · Authors · 2021-11-18
> **Response to Reviewer 4v5L - Part 2**
>
> > The loss function choice seems heuristic and not very well explained. How does the loss help with periodicity? What happens if the signals have high vs. low frequency components? What happens when the signals have peaks? Different impactful time-series cases should be considered and the section should be expanded accordingly.
>
> We thank the reviewer for this comment. Our loss function is indeed heuristic. We would like to point out that the choice of loss function is not the main contribution of our paper and we opted for losses that were proposed in literature. Our main motivation was to use the LSGAN loss because it has been shown to address mode collapse (Mao et al. 2017) and that least-square-like losses (albeit in embedding space) have been used for time series GANs before (Morgen 2016, NeurIPS Constructive Machine Learning Workshop; Yoon et al. 2019, NeurIPS). Although the LSGAN loss alone worked quite well, we found that sometimes the moments between synthetic and real samples were not aligned. Thus, we have added the moment loss as an additional auxiliary loss and show its effectiveness in the ablation (previously Figure 7, now Figure 6). We revised the Loss Function section on p. 4. to reflect our motivation.
> We thank the reviewer for the suggestion to study high vs. low frequency components and peaks in time series data. However, we think that such a study (which would also require the consideration of different loss functions) goes beyond the scope of our work. To the best of our knowledge, such a study was not carried out yet and the loss functions of recent time series GAN models are based on heuristics. We agree that investigating the loss function choice more closely in future studies will be useful for the field.
>
> > I don't think the FID score would have significant value for time-series as the perceptual quality of time-series to humans is different than images. Beyond proposing the score, its usefulness should be demonstrated.
>
> We thank the reviewer for this comment. We are convinced that the Context-FID score is a main contribution of our paper because GAN evaluation in the time series domain is not well studied. Note that unlike the standard FID formulation in the image field, we use an unsupervised embedding network (Franceschi et al. 2019, NeurIPS) to compute time series embeddings of real and fake samples between which we then compute the Wasserstein-2 metric. Thus, we use a very different embedding network than image FID scores do that is geared towards time series. In our downstream forecasting tasks, we are particularly interested in replacing missing data with synthetic time series samples. Thus, it is important that the synthetic sample “fits” into the local time series context. This is achieved with the embedding network that learns the in-context fit via contrastive learning (Franceschi et al. 2019, NeurIPS). We explain this motivation on p. 4 of the manuscript in the beginning of the Experiments section.
> To showcase the usefulness of our approach, we find that the best models by Context-FID also perform well in the downstream forecasting tasks. We have performed additional analysis and updated the paper to show that the Context-FID score correlates with the forecasting performance (NRMSE) of the GAN model (p. 9 and Table 4 in Appendix). We think this provides sufficient evidence that the Context-FID score is useful for developing GAN models in the time series domain.
>
> > The procedure for hyperparameter tuning is not well explained. How does one construct a validation set and what validation reward to use?
>
> We thank the reviewer for this comment. Indeed, we did not explain the hyperparameter tuning and added this to the Appendix (Section D/Hyperparameter Tuning). Our goal was to find a good set of default hyperparameters that perform well for all datasets. We did a limited grid-search by tuning the number of sampled batches per epoch, the batch size, and enabled/disable self-attention and progressive layer fade-in (details are added to the paper). We selected the hyperparameter set with the lowest Context-FID on the training set, averaged over the four datasets. We use the same hyperparameter set when reporting Context-FID on all datasets and also for the downstream forecasting tasks.
> We added a correlation analysis that shows the Context-FID on the training set correlates well with the forecasting performance of PSA-GAN on a hold-out set (Table 4 in Appendix). This data suggests that the described hyperparameter setting approach does not lead to overfitting on the training set.

---

> ### Author Response · Authors · 2021-11-18
> **Response to Reviewer 4v5L - Part 3**
>
> > I think experiments section should be better organized and the writing should be improved for a more clear presentation. The results on far-forecasting and data imputation seem quite promising. The improvements in cold start and data augmentation seem negligible. For forecasting, the gap with DeepAR is still significant. Overall, rather than focusing on a lot of results, I suggest emphasizing on the key capabilities the data synthesis brings, with significant improvements. Fig. 6 is just a cherry-picked example and it does not tell much.
>
> We thank the reviewer for this comment. We agree that the experiments section could be more structured and distinguish the forecasting tasks where the synthetic data brings significant improvements and where the improvements are less clear. However, we think it is still important to report the results of all the downstream forecasting task experiments because they introduce a way of evaluating GAN models for future work. Furthermore, we think it is important to inform the community about the negative outcome of our data augmentation study.
>
> For forecasting, we would like to point out that DeepAR does only outperform PSA-GAN because of the lagged inputs which we do not use for PSA-GAN. When we use the same context window of 64 time steps in both models, the PSA-GAN-C variant (with context) actually performs better. PSA-GAN (which does not use context) is still on par with DeepAR-C (which uses 64 time steps as context). Thus, the data suggests that PSA-GAN does perform well in forecasting, even though it was not developed to be used as a forecasting model (and we do not believe that this is the primary purpose of this model). We updated the Forecasting section on p. 8 to clarify this.
> Regarding Fig. 6, we still think it is important to show the samples that our model generates but we agree with the reviewer that this does not have to be in the Experiments section. We removed the Figure and pointed to the sample plots in the Appendix (Section: Sample Forecasts).
>
> > For data augmentation, how can one optimize what ratio of synthetic data to use and is there a way of more intelligent sampling of synthetic data specifically focusing on the low data coverage regimes? The naive application does not seem promising and the authors need to think further to show strong results.
>
> We thank the reviewer for this comment. As pointed out in the paper, we think that DeepAR might be able to model the specific datasets in our study quite well already and data augmentation does not bring additional improvements, but instead adds unnecessary noise. We actually experimented quite extensively with ways of using the synthetic data in data augmentation, including: testing different ratios of real and synthetic samples, testing different weightings when averaging real and synthetic samples and testing different sampling schedules. However, all these approaches did not perform better than the naive approach presented in the paper. The suggestion of the reviewer to focus on low data coverage regimes is interesting, but the datasets used in this paper have uniform coverage, so it is not clear how to achieve this in this paper. We would like to point out that other studies also find that improving downstream tasks with GAN data augmentation does not work well (see Ravuri and Vinyals 2019, ICLR Workshop Learning from Limited Labeled Data). We still believe that we should report this (negative) result.

---

> ### Author Response · Authors · 2021-11-18
> **Response to Reviewer 4v5L - Part 4**
>
> > Many recent papers show significant outperformance over DeepAR. That aspect is orthogonal to the contributions of the paper but I suggest benchmarking with at least one more forecasting model. Specifically for far-forecasting, RNN based models are known to suffer from error propagation without teacher forcing. So direct decoding approaches that generate all the multi-horizon forecasts would be better candidates, and likely will reduce the gains with synthetic data.
>
> We thank the reviewer for this comment and we agree that the paper would benefit from additional benchmarking with other forecasting models. To this end, we have implemented our setup with NBEATS (Oreshkin el al. 2020, ICLR) and Temporal Fusion Transformer (TFT) (https://arxiv.org/abs/1912.09363, to appear in the International Journal of Forecasting). As the reviewer suggests, NBEATS generates multi-horizon forecasts and should not suffer from error propagation induced by teacher forcing. Below, we provide the results we generated so far for the far-forecasting experiment. Overall, we find that NBEATS and TFT benefit to a similar degree from the synthetic time series data as DeepAR (with the exception of the Solar dataset that NBEATS seem not to be able to model well).  Note that our NBEATS implementation only uses an ensemble of three neural networks (instead of the original authors 180 neural networks) due to the long runtime of training 180 neural networks. At the time of writing this comment, we are generating the results for the other downstream tasks and hope that we can provide the results by the end of the discussion period.
>
> | model   | (NRMSE, Electricity)   | (NRMSE, M4)   | (NRMSE, Solar)   | (NRMSE, Traffic)   |
> |:--------|:---------------------------|:-------------------------|:----------------------------|:-----------------------|
> | NBEATS  | 5.29 +\- 0.28              | 1.92 +\- 0.04            | 2.08 +\- 0.0                | 0.94 +\- 0.02               |
> | EBGAN      | 3.59 +\- 0.21              | 1.76 +\- 0.03            | 2.07 +\- 0.0                | 0.96 +\- 0.0           |    |
> | TIMEGAN      | 2.06 +\- 0.02              | 1.75 +\- 0.01            | 2.07 +\- 0.0                | 0.8 +\- 0.01           |
> | PSA-GAN (ours)      | 1.3 +\- 0.04               | 0.4 +\- 0.01             | 2.07 +\- 0.0                | 0.65 +\- 0.0           |
>
>
> | model   | (NRMSE, Electricity)   | (NRMSE, M4)   | (NRMSE, Solar)   | (NRMSE, Traffic)   |
> |:--------|:---------------------------|:-------------------------|:----------------------------|:-----------------------|
> | TFT     | 4.51 +\- 0.02              | 1.78 +\- 0.04            | 1.74 +\- 0.02               | 0.94 +\- 0.0           |
> | EBGAN      | 3.54 +\- 0.22              | 1.5 +\- 0.05             | 2.5 +\- 0.13                | 0.76 +\- 0.02          |
> | TIMEGAN      | 2.03 +\- 0.05              | 1.71 +\- 0.04            | 1.26 +\- 0.03               | 0.82 +\- 0.03          |
> | PSA-GAN (ours)      | 1.97 +\- 0.05              | 0.34 +\- 0.01            | 2.15 +\- 0.13               | 0.7 +\- 0.0            |

---

> ### Author Response · Authors · 2021-11-22
> **Reponse to Reviewer 4v5L - Part 5**
>
> We sincerely thank the reviewer for the response and for updating the score. We focused on performing the additional requested experiments in the rebuttal period and have edited the paper to add these additional experiments and provide further clarification.
>
> We have added the full set of experiments with NBEATS and TFT as forecasting models to the paper. Overall, we find that the proposed DeepAR/PSA-GAN combination performs best in the downstream tasks (lowest absolute NRMSE). Therefore, we discuss this combination in the main text and have added the additional results to the Appendix.
>
> Also for NBEATS and TFT, we find that using PSA-GAN samples leads to the largest NRMSE improvements in the far-forecasting and missing value imputation experiments. For cold starts, the results are mixed. For data augmentation, no GAN model is able to consistently improve over NBEATS and TFT. One interesting observation is that the gap between the TFT/PSA-GAN and the TFT/TIMEGAN combination is much smaller than for DeepAR/PSA-GAN and DeepAR/TIMEGAN. We hypothesize that a RNN-based and an attention-based architecture might complement each other in this forecaster/GAN setup. However, the DeepAR/PSA-GAN architecture still performs best overall.

---

### Official Review · Reviewer_RLDM · 2021-11-02

**Correctness:** 4
**Technical Novelty And Significance:** 3
**Empirical Novelty And Significance:** 3
**Recommendation:** 6
**Confidence:** 3

**Main Review:**

Positives:
1. The proposed approach is novel for generating time series data.
2. The paper introduces a metric for evaluating the quality of synthetic time series data.
3. This paper provides comprehensive experiments, including both qualitative analysis and quantitative results, to show the effectiveness of the proposed framework.
4. The paper is well written and easy to follow.

Concerns:
1. Could the authors clarify how the loss function in Equation 7 is used to train the generator and the discriminator?
2. What are the time features (referred as X in the paper)?
3. How does the concatenation happen in the generator with the time features since the sequence length after upscaling keeps increasing?
4. Similarly, the authors should provide more details on how does the approach deals with the fixed-length context in PSA-GAN-C? How does the summation work with varying lengths of sequences?
5. Table 3 compares the forecasting performance of GAN models with the DeepAR architecture. It shows that DeepAR achieves much better performance. Based on this experiment, wouldn't it be better to use DeepAR for imputing/predicting the time series in the far-forecasting experiment than imputing with the GAN? The authors should compare to this baseline or clarify why this is not possible.
6. Table 2 shows interesting results. Could the authors explain why the data augmentation leads to degradation of the forecasting performance instead of improving it?


Additional Comments:
1. The discriminator architecture in Figure 1 and equation 5 don't agree with each other. Could the authors clarify/align them?
2. Remove the additional 'of' in section 4 para 1.

**Summary Of The Paper:**

This paper proposes a GAN-based approach (PSA) for generating realistic time-series data that can be used to improve several downstream tasks such as imputation of missing data and forecasting. The generator and discrimination model in the proposed GAN framework, PSA-GAN consists of progressively growing blocks consisting of convolutions and self-attention. The paper also proposes a metric for evaluating the quality of synthetic time series data. Experiments show that the proposed approach is able to achieve better performance than previous GAN-based approaches for time series and is also able to improve on downstream forecasting tasks by augmenting forecasting models.

**Summary Of The Review:**

This paper proposes a novel approach for generating realistic time-series data. It also introduces a metric for evaluating the quality of synthetic time series data which has been lacking in this domain. The paper also provides comprehensive experiments to show the effectiveness of the proposed framework.

---

> ### Author Response · Authors · 2021-11-18
> **Reponse to Reviewer RDLM - Part 1**
>
> We thank the reviewer for the thorough review of our paper. We address individual concerns below.
>
> > Could the authors clarify how the loss function in Equation 7 is used to train the generator and the discriminator?
>
> We revised section 3.1 to add some clarity. The loss of Eq. (7) is an auxiliary loss to the generator. It works as follows:
> Given a batch of real samples Z_{t, \tau} we compute the time features X_{t, \tau}. Using X_{t, \tau} we compute a batch of synthetic samples \hat{Z}_{t, \tau}. We then compute the unsupervised loss of the generator L_G and the moment loss ML(Z_{t, \tau}, \hat{Z}_{t, \tau}). Lastly, we backpropagate the sum of those losses through the generator.
>
> > What are the time features (referred as X in the paper)?
>
> Thank you for this question. The time features X we have considered correspond to: 1) hour, 2) day of the week, 3) day of the month, 4) day of the year, and 5) age feature. These features are further described in Appendix A (p.12) and we revised to the section on p. 2 to make it more clear.
>
> > How does the concatenation happen in the generator with the time features since the sequence length after upscaling keeps increasing?
>
> Thank you for raising this question. At each step the time features are downscaled to match the size of the current sequence length. As an example, for the case where the target length is 64=2**6, at the first step the time features will be downscaled to a length of eight through average pooling and concatenated to the noise vector. At the second step, the sequence will be of length 16. Therefore the original time features of length 64 will be downscaled this time to 16 and concatenated to the current sequence. Next step they will be concatenated to length 32 and concatenated and so on.
>
> > Similarly, the authors should provide more details on how does the approach deals with the fixed-length context in PSA-GAN-C? How does the summation work with varying lengths of sequences?
>
> We thank the reviewer for this question. In our approach the length of the context is fixed to be 64. We concatenate the context vector along the feature dimension, i.e. each time step of the sequence is concatenated to the context vector. Observe that as the time dimension of the context is fixed it is not possible to concatenate the context vector to the current sequence along the time dimension. We revised the section “Model/PSA-GAN-C” to clarify this topic.
>
> > Table 3 compares the forecasting performance of GAN models with the DeepAR architecture. It shows that DeepAR achieves much better performance. Based on this experiment, wouldn't it be better to use DeepAR for imputing/predicting the time series in the far-forecasting experiment than imputing with the GAN? The authors should compare to this baseline or clarify why this is not possible.
>
> We thank the reviewer for this comment. We would like to point out that this experiment doubles as a far-forecasting experiment and missing value presence during inference. Note that DeepAR imputes the missing values of its imminent context in a principled way by filling missing values from drawn samples during unrolling. This can be seen in Figure 2 where the GAN does not bring any benefit over DeepAR for the windows 2 and 3 (where values can still be imputed by DeepAR), which is consistent with the suggestion of the author. However, as we shift the backtest data (or viewing this as having longer missing value stretches before inference), DeepAR’s performance deteriorates because the model is not able to impute the lagged values. In this case, filling the lagged values with samples from the GAN models improves the performance.
>
> Technically, an alternative way to approach this could be to re-train the DeepAR model with a larger prediction length and shift the forecast data before the missing values are starting. However, the computational cost and complexity of the model increases with the increasing prediction length. The approach that we showcase in our manuscript does not require re-training and allows for reacting to suddenly appearing missing values during inference time on-the-fly, which might be useful if re-training a larger model would take too long for a given application.

---

> > ### Comment · Reviewer_RLDM · 2021-11-29
> > **Response follow up**
> >
> > I would like to thank the authors for addressing my concerns. I'm happy with the current state of the paper and leaning towards acceptance.

---

> ### Author Response · Authors · 2021-11-18
> **Response to Reviewer RLDM - Part 2**
>
> > Table 2 shows interesting results. Could the authors explain why the data augmentation leads to degradation of the forecasting performance instead of improving it?
>
> We thank the reviewer for this comment. As pointed out in the paper, we think that DeepAR might be able to model the specific datasets in our study quite well already and data augmentation does not bring additional improvements, but instead adds unnecessary noise. We actually experimented quite extensively with ways of using the synthetic data in data augmentation, including: testing different ratios of real and synthetic samples, testing different weightings when averaging real and synthetic samples and testing different sampling schedules. However, all these approaches did not perform better than the naive approach presented in the paper. The suggestion of the reviewer pbaT to focus on low data coverage regimes is interesting, but the datasets used in this paper have uniform coverage, so it is not clear how to achieve this in this paper. We would like to point out that other studies also find that improving downstream tasks with GAN data augmentation does not work well (see Ravuri and Vinyals 2019, ICLR Workshop Learning from Limited Labeled Data). We still believe that we should report this (negative) result.
>
> > The discriminator architecture in Figure 1 and equation 5 don't agree with each other. Could the authors clarify/align them?
>
> Thank you for detecting this inconsistency. We have revised the section  “Model/PSA-GAN/Discriminator” (p.4) to resolve this issue:
> we corrected the output of the generator in Figure 1 to be of length 2^(n+3)
> in Eq. (5) we re-indexed the set of functions $d_i$ from the range [L, 2] to the range [L+1, 2]. In the former case the first module (d_{L+1}) was missing to accurately describe our model
> in Eq. (4) we re-named the function $d_{L+1}$ to be $d^{‘}_{L+1}$ to avoid a notational collision with Eq. (5)
>
> > Remove the additional 'of' in section 4 para 1.
>
> Thank you for your comment. We have updated the paper accordingly.

---

### Official Review · Reviewer_pbaT · 2021-11-03

**Correctness:** 4
**Technical Novelty And Significance:** 3
**Empirical Novelty And Significance:** 3
**Recommendation:** 6
**Confidence:** 3

**Main Review:**

### Strengths
1. The proposed GAN-model seems to outperform the baselines significantly.
2.  Adapting the FID score to time series seems like a good idea. This could be an interesting measure of performance for future work.

### Weaknesses
There are multiple issues in the description of the model which are unclear to me. I think that Chapter 3 (and Figure 1) would benefit from being partially rewritten.
1. How is $\phi(i)$ used in the model?
2. What is meant by the time features $\mathbf{X}$? Do these features encode the position in time, like the e.g. sine-cosine in the transformer? What does the “number of time features” mean? Is this the dimensionality of the feature vector at each time?
3. Page 3 says: “As a preprocessing step, we first map the input features from a sequence of length τ to a sequence of length 8, denoted by $\tilde{Z}_0$, using average pooling.” What is meant by the input features? Is that the sum $\mathbf{X} + \mathbf{n}$?
4. What is the dimensionality of the Gaussian noise $\mathbf{n}$?
5. According to Figure 1, the output of the generator has length $n * 16$, where $n$ is the number of blocks. Since each block doubles the size of its input, shouldn’t the output have size $2^n * 8$?
6. Shouldn’t the output of equation (4) be named $Y_L$ instead of $\tilde{Z}_L$, since the latter denotes a variable inside the generator?
7. p. 4: The time range of the $\hat{Z}_{i, t-L_C, L_C}$ is said to be $[t-L_C, t-1]$, but according the definition of $\hat{Z}_{i, t, \tau}$ on p. 2, the time range would be $[t-L_C, t]$.

**Technical Novelty And Significance:**

* Context-FID: (1) Why is the encoder of Franceschi used - comparison to other encoders or why is it particularly good? (2) Why is the network trained separately for each dataset? This is different to the standard FID, where the Inception network is trained on ImageNet.
Minor:
* Page 3 says: “The module $m: f(x) \rightarrow m(f(x))$ is referenced as Residual Self-Attention in Fig. 1 (Right).” This module should not be denoted by $m$ because the variable m already denotes the main function defined in equation (1). So the sentence should be something like: “The module $m_1: f(x) \mapsto m(x)$ …”.

**Typos**
* “The discriminator contains n blocks that halves the size of the input using an average pooling.” -> The discriminator contains n blocks that *halve* the size the size of the input *using average* pooling.
* “The training procedure to address those issues are presented in the appendix.” -> *is* presented in the appendix


**Summary Of The Paper:**

The paper proposes a type of GAN to generate synthetic time series. The authors use the data generated by their proposed model to train forecasting networks, which improves the baselines. They also show that the proposed GAN can itself be used as  a forecasting model and that it has competitive performance to baseline models. Finally, the paper proposes an adaptation of the FID score for time series.

**Summary Of The Review:**

The proposed model outperforms the baselines and the proposed FID score for time series looks interesting. However, there are a few unclarities in the description of the model. I hope that these things can be explained during the rebuttal.

---

> ### Author Response · Authors · 2021-11-18
> **Response to Reviewer pbaT**
>
> We thank the reviewer for the detailed comments. We address the concerns below.
>
> > How is phi(i) used in the model?
>
> Thank you for your comment. Phi(i) is an embedding network that maps the index of a time series to a fixed size vector that is concatenated to each time step of the input. We revised section 3, “Problem formulation” (p. 2) and Appendix A (p.12) to clarify the use of features.
>
> > What is meant by the time features X? Do these features encode the position in time, like the e.g. sine-cosine in the transformer? What does the “number of time features” mean? Is this the dimensionality of the feature vector at each time?
>
> Thank you for your comment. The time features X we have considered correspond to: 1) hour, 2) day of the week, 3) day of the month, 4) day of the year, and 5) age feature. These features are further described in Appendix A (p.12)
> The “number of time features”: it is the dimensionality of the time features at each time step, i.e. D= 5. Further clarity has been brought in the revised paper in section 3, “Problem formulation” (p. 2).
>
> > Page 3 says: “As a preprocessing step, we first map the input features from a sequence of length τ to a sequence of length 8, denoted by Z˜ 0, using average pooling.” What is meant by the input features? Is that the sum X + n?
>
> Thank you for this comment. By input features we mean: [gaussian noise, phi(i), time features] which has dimension $\tau \times 16$, where gaussian noise is of dimension $\tau \times 1$, phi(i) is of dimension $\tau \times 10$, and the time features considered are of size $\tau \times D$. We have added further details to Section 3/PSAGAN/Generator to make this clearer.
>
> > What is the dimensionality of the Gaussian noise n?
>
> Thank you for this question. The dimensionality of the Gaussian noise n is $\tau \times 1$. We have clarified this in section 3, “Problem formulation” (p. 2).
>
> > According to Figure 1, the output of the generator has length n * 16, where n is the number of blocks. Since each block doubles the size of its input, shouldn’t the output have size 2^n * 8?
>
> Thank you for pointing out this inconsistency. We have corrected Fig. 1 and corrected its output to the value of 2^(n+3) which equals  2^n * 8  as 2^(n+3)=2^n * 8
>
> > Shouldn’t the output of equation (4) be named Y_L instead of \tilde{Z}L, since the latter denotes a variable inside the generator?
>
> Thank you for this comment. The observation you have made is correct. We have updated the paper with the output name of equation (4) to be Y_L.
>
> > p. 4: The time range of the \hat{Z}{i, t-L_C, L_C} is said to be [t-L_C, t-1], but according the definition of \hat{Z}_{i, t, \tau} on p. 2, the time range would be [t-L_C, t].
>
> Thank you for spotting this. We have changed the definition of \hat{Z}_{i, t, \tau} to be (z_{i, t}, …, z_{i, t + tau - 1}) to be consistent across the paper.
>
> > Why is the encoder of Franceschi used - comparison to other encoders or why is it particularly good?
>
> We thank the reviewer for this question. The motivation and advantages that we find to use this specific encoder are as follows:
> It embeds varying length time series into a fixed size vector. This is a desirable property for us as we train models for different lengths and in general imposes no constraints on the size of the time series.
> It makes no structural assumption on time series.
> It’s a general purpose embedding that has been confirmed by extensive experiments by Franceschi et al.
> It is shown to be scalable and thus can provide meaningful embedding for short and long time series.
>
> However, we believe that other encoder choices could achieve the same advantages. We have selected this encoder because it has been shown by Franceschi et al. to work well in the time series embedding tasks and our early experiments confirmed this. We added an additional analysis (Table 4 in Appendix) and show that the Context-FID score correlates with the forecasting performance of the GAN model. This data supports the use of this encoder in the time series domain.
>
> > Why is the network trained separately for each dataset? This is different to the standard FID, where the Inception network is trained on ImageNet.
>
> Thanks for your comment. In our paper, we followed the original paper of Franceschi where they trained an encoder per dataset. For image data, deep learning models might extract low-level features (like edges) that are shared between image datasets and therefore retraining on a new dataset might not be necessary. In the time series domain, datasets can differ in terms of volatility, seasonality and granularity. Therefore, it is not clear whether common low-level features can be compared between datasets and this would be an interesting future research direction, particularly for meta learning in time series.. Training one embedding network per dataset ensures that dataset-specific features are captured.
>
> We also addressed the minor comments and thank the reviewer for pointing them out.

---

> ### Author Response · Authors · 2021-11-23
> **Response to Reviewer pbaT - Part 2**
>
> We thank the reviewer for taking the time reviewing our work.
>
> We would like to understand why the reviewer gave score 2 for correctness (2: Several of the paper’s claims are incorrect or not well-supported).
>
> Which claims are incorrect and/or not well-supported in the view of the reviewer? The majority of the review was on notation (which we addressed in the response) and about the encoder choice training (which we addressed as well). We are happy to work with the reviewer to improve the paper if there are still concerns about claims that are not well-supported.

---

> > ### Comment · Reviewer_pbaT · 2021-11-25
> > **Reply**
> >
> > My low score for correctness was due to the inclarities that I had raised. Since the authors have addressed these issues, I have raised the score to 4.
> >
> > After reading the replies of the authors and the other reviews, I have decided to keep my overall score as it is.

---

### Official Review · Reviewer_DRP7 · 2021-11-04

**Correctness:** 4
**Technical Novelty And Significance:** 2
**Empirical Novelty And Significance:** 3
**Recommendation:** 6
**Confidence:** 5

**Main Review:**

The idea of progressive growing of GANs is convincing for synthesizing time series. The time series data is difficult to fit due to high volatility. Thus it is feasible to fit the distribution gradually from easy case to difficult case. Self-attention is capable of promoting the capability of fitting abnormal circumstances such as data imputation. So, the idea is convincing.

But there are two drawbacks for this submission.
1) The novelty is not good enough for ICLR. The two ideas are both borrowed from existing papers.
2) The SOTA algorithms are not compared except TimeGAN, such as QuantGAN, SigCWGAN, and DAT-GAN. EBGAN is not usually applied  in this scenario. Besides, it is not the SOTA algorithm in the GAN field.



**Summary Of The Paper:**

The authors propose a new GAN-based algorithm for time series synthesis. They use progressive growing of GAN architectures to improve the performance of GAN and self-attention to enhance the expressive capability of neural networks. The experimental results validate the superiority of the proposed PSA-GAN algorithm.

**Summary Of The Review:**

The idea is interesting. But the novelty and experiments are not good enough for ICLR.

---

> ### Author Response · Authors · 2021-11-18
> **Reponse to Reviewer DRP7**
>
> We thank the reviewer for the comments. We address individual concerns below.
>
> > The novelty is not good enough for ICLR. The two ideas are both borrowed from existing papers.
>
> We thank the reviewer for this comment. We agree that the components of the PSA-GAN model (progressive growing, self-attention, least-squares loss, moment loss)  have been proposed and studied before. However, we believe that this combination is non-trivial and has not been proposed in the time series domain, yet. Additionally, our paper makes contributions on the non-trivial problem on how to evaluate GAN models in the time series domain. To this end, we propose the Context-FID score based on unsupervised time series embeddings. We have performed additional analysis and updated the paper to show that the Context-FID score correlates with the forecasting performance (NRMSE) of the GAN model (p.9). Additionally, we evaluate the GAN models by showing their added value in four challenging forecasting scenarios (missing values during inference/far-forecasting, missing value stretches, cold-starts, data augmentation), thus providing practitioners with a blue-print on how to leverage time series GAN models. Overall, we think that our paper makes contributions beyond the PSA-GAN model. Specifically, our contributions on GAN model evaluation and their use in forecasting are highly relevant and have not been studied extensively in the time series domain.
> > The SOTA algorithms are not compared except TimeGAN, such as QuantGAN, SigCWGAN, and DAT-GAN. EBGAN is not usually applied in this scenario. Besides, it is not the SOTA algorithm in the GAN field.
>
> We thank reviewer DRP7 for the references. We have added the references to the literature review and discuss them accordingly in Section 2/Related Work/p.2
>
> The references provided are indeed interesting and belong to the specialized field of financial time series where the main challenges, as mentioned in a recent survey (A), can be summarized as stylistic factors, namely: 1) absence of autocorrelations, 2) heavy tails, 3) volatility clustering (high volatility events tend to cluster together), 4) Gain/loss asymmetry, and 5) aggregational Gaussianity (the distribution of returns asymptotically converges to a Gaussian distribution on time).
>
> The references provided by the reviewer are mainly inspired, and designed, by these challenges. In particular: QuantGAN (B) aims at approximating a realistic asset price simulated by using neural networks; DAT-GAN(C) is suitable for the context of financial portfolio management; and SigCWGAN(D) is numerically evaluated on the task of log return of the close prices and the log of median realised volatility of the S\&P 500 index (SPX) and Dow Jones index (DJI).
>
> Whereas these challenges are clearly relevant we believe that this line of work (designing suitable loss functions for financial time series) is orthogonal to the challenges of this paper (designing suitable GAN architectures for time series and focusing on the evaluation and downstream impact of GANs). Moreover, we have provided a novel Context-FID score that serves as a proxy to estimate the effectiveness of GANs in the context of time series forecasting, and show that it is highly correlated to the performance observed in our experiments. We think it would be interesting to combine these ideas (loss functions) with our work (an attention-based time series GAN architecture) in future work.
>
> Despite the different focus and goals of GANs for financial time series, and ours focused on time series forecasting, we have tried to compare with the proposed related work. We contacted the authors of QuantGAN asking for any available code but have not received the requested code. For SigCWGAN we have taken the code available at https://github.com/SigCGANs/Conditional-Sig-Wasserstein-GANs, but were not able to run it successfully. For DAT-GAN there is no freely available code. Implementing these methods with adequate benchmarking is not feasible during the discussion period.
>
> References:
>
> A: Florian Eckerli and Joerg Osterrieder. Generative Adversarial Networks in Finance: An Overview. Machine Learning eJournal. 2021.
>
> B: Magnus Wiese and Robert Knobloch and Ralf Korn and Peter Kretschmer. Quant GANs: deep generation of financial time series. Quantitative Finance. 2020
>
> C: Sun, He and Deng, Zhun and Chen, Hui and Parkes, David C. Decision-Aware Conditional GANs for Time Series Data. arXiv. 2020
>
> D: Ni, Hao and Szpruch, Lukasz and Wiese, Magnus and Liao, Shujian and Xiao, Baoren. Conditional Sig-Wasserstein GANs for Time Series Generation. arXiv. 2020

---

> > ### Comment · Reviewer_DRP7 · 2021-11-26
> > **Thanks for the detailed reply**
> >
> > Your interpretation about other SOTAs is convincing. And the difficulty of acquiring associated open source makes extra experiments on other SOTAs infeasible during the discussion period. I also read other reviewers' comments and your replies. You treat your work seriously enough to get an acceptance score.

---

### Decision · Program_Chairs · 2022-01-20

**Decision:**

Accept (Poster)

**Comment:**

This paper adapts the idea of progressive growing of GANs to time series synthesis. The reviewers thought that the idea was well motivated. DRP7 initially expressed concern w.r.t. novelty. They were also concerned with the lack of certain baselines. The authors responded, highlighting its contributions w.r.t. Evaluation (Context-FID score) and extensiveness of the evaluation. The authors also added missing references but pushed back on the additional baselines. DRP7 raised their score.  Reviewer pbaT was also positive about the work though had some questions and suggestions for improving clarity. They had initially given a low score for “correctness” but raised this, indicating they were satisfied their clarity concerns were addressed. Reviewer RLDM (whose code-name happens to match a ML conference) thought the work was novel and appreciated the introduction of a new metric for evaluating the quality of generated time series data. They remarked on the thoroughness of the experiments and the quality of the presentation. They asked some clarifying questions to which the authors provided a response. Reviewer 4v5L also had a concern with novelty, felt the loss function was “heuristic” and didn’t see the utility of the FID-based score. They also presented several clarifying questions. The authors provided a lengthy response to that reviewer’s concerns, having run additional analysis, and the reviewer upgraded their score in response. With all reviewers on the accept side of the fence I am inclined to recommend acceptance. Please note 4v5L’s comment that “the paper still needs significant edits to reflect the points in the reviewer responses”.